# ATPO: Adaptive Tree Policy Optimization for Multi-Turn Medical Dialogue

**Ruike Cao[1,2,*]   Shaojie Bai[2,3,*]   Fugen Yao[2,†]   Liang Dong[2]   Jian Xu[2]   Li Xiao[1]**
[1]University of Science and Technology of China
[2]Qwen Applications Business Group, Alibaba Group
[3]Zhejiang University

## Abstract

Effective information seeking in multi-turn medical dialogues is critical for accurate diagnosis, especially when dealing with incomplete information. Aligning Large Language Models (LLMs) for these interactive scenarios is challenging due to the uncertainty inherent in user-agent interactions, which we formulate as a Hierarchical Markov Decision Process (H-MDP). While conventional Reinforcement Learning (RL) methods like Group Relative Policy Optimization (GRPO) struggle with long-horizon credit assignment and Proximal Policy Optimization (PPO) suffers from unstable value estimation in this context, we propose a novel uncertainty-aware **A**daptive **T**ree **P**olicy **O**ptimization (**ATPO**) algorithm. Our method adaptively allocates the rollout budget to states with high uncertainty, quantified by a composite metric of Bellman error and action-value variance. This strategy enables more accurate value estimation, while fostering more efficient and diverse exploration. To mitigate the high computational cost of tree-based RL, we introduce two key optimizations: an uncertainty-guided pruning mechanism to minimize the number of rollouts, and an asynchronous search architecture that leverages KV cache reuse to maximize inference throughput. Extensive experiments on three public medical dialogue benchmarks demonstrate that our algorithm significantly outperforms several strong baselines, culminating in Qwen3-8B model surpassing the much larger GPT-4o ($+0.92\%$ accuracy). [1]

## 1 Introduction

In recent years, Large Language Models (LLMs) such as GPT-4 Achiam et al. (2023), Gemini 2.5 Comanici et al. (2025), Qwen3 Yang et al. (2025), and DeepSeek-R1 Guo et al. (2025a) have demonstrated exceptional capabilities across a range of natural language processing tasks, including open-domain question answering, dialogue generation, and code generation, continuously pushing the boundaries of AI performance (Chen et al., 2025). These models are increasingly being applied to downstream domains like education (Chu et al., 2025), law (Siino et al., 2025), and healthcare (Awasthi et al., 2025). Within the medical field, medical LLMs are consistently achieving state-of-the-art results on various benchmarks (Sellergren et al., 2025; Li et al., 2025a), such as professional medical examinations (Jiang et al., 2025) and disease diagnosis tasks (Xu et al., 2025; McDuff et al., 2025), and show immense potential for providing preliminary medical advice and assisting in clinical decision-making (Kopka et al., 2025).

However, despite these achievements, a critical aspect has long been overlooked. Current training and evaluation of medical LLMs predominantly focus on single-turn interaction scenarios, where models are expected to provide faithful responses based on the user's initial input. In real-world medical dialogues, however, the information provided by users is often incomplete, making it difficult to generate a satisfactory response based solely on the vague or fragmented initial query (Auerbach et al., 2024; Wu et al., 2025a). This necessitates the model's ability to proactively ask clarifying

---

[*]Equal contribution.
[†]Corresponding author: fugen.yfg@alibaba-inc.com.
[1]Code: https://github.com/Quark-Medical/ATPO.

questions to gather more essential information. Unfortunately, this capability for dynamic, multi-turn information gathering is a significant deficiency in current models (Laban et al., 2025).

Previous work has attempted to fill this gap. Some approaches Li et al. (2024); Liu et al. (2025a); Hu et al. (2024) have used prompt engineering to elicit proactive questioning, but these methods often fail to fundamentally enhance the model's multi-turn interactive capabilities and can even lead to poorer performance than simply responding with incomplete information. Other efforts Liao et al. (2023); Liu et al. (2025b) have employed supervised fine-tuning (SFT) to improve dynamic interaction, yet these models tend to merely imitate the training data. Furthermore, some studies Shi et al. (2024); Xiong et al. (2024) have extended single-turn preference optimization to the trajectory level, but they rely on costly preference data and are highly sensitive to distribution shift. While reinforcement learning offers a promising, goal-driven alternative with stronger generalization (Guo et al., 2025a), current methods are also flawed. For instance, Group Relative Policy Optimization Shao et al. (2024) struggles with long-horizon credit assignment, and Proximal Policy Optimization Schulman et al. (2017) often suffers from unstable value estimation, hindering effective policy learning for complex multi-turn dialogues (Feng et al., 2025a).

In this work, we introduce Adaptive Tree Policy Optimization, a novel uncertainty-aware algorithm illustrated in Figure 1. ATPO employs an adaptive tree search where, for each node, it calculates an uncertainty metric to decide whether to expand the search further. This metric is a composite of two key signals: the Bellman error, which prioritizes nodes beneficial for critic training, and the action-value variance, which encourages sampling diversity. Furthermore, ATPO achieves high training efficiency by reusing shared prefixes to fully leverage the Key-Value (KV) cache Kwon et al. (2023) mechanism, combined with an asynchronous execution strategy. We conduct comprehensive evaluations on three Qwen3 models Yang et al. (2025) with different sizes (Qwen3 1.7B, 4B and 8B) using three multi-turn medical dialogue datasets meticulously adapted from public multiple-choice question datasets (MedQA Jin et al. (2020), MedMCQA Pal et al. (2022), and MedicalExam Liao et al. (2024)). The experimental results demonstrate that our proposed method significantly outperforms strong competitive RL baselines across all datasets and model sizes, validating its effectiveness and generalization capabilities.

Our contributions are as follows:

- We propose the Adaptive Tree Policy Optimization algorithm, which adaptively allocates rollout budgets based on turn-level uncertainty in multi-turn medical dialogues. This method enhances sampling diversity while simultaneously improving the critic model's accuracy.

- We design ATPO to be highly efficient by reusing shared prefixes to fully leverage the KV cache, and we implement it with an asynchronous execution strategy to achieve substantial improvements in inference throughput.

- Extensive experiments demonstrate that ATPO not only consistently and significantly outperforms strong RL baselines, but also achieves this with far greater sample efficiency, validating its robust generalization and effectiveness.

## 2 RELATED WORK

### 2.1 REINFORCEMENT LEARNING IN MULTI-TURN MEDICAL DIALOGUES

Recent efforts have applied reinforcement learning to train medical LLMs for proactive, multi-turn dialogue. A common approach involves simulating doctor-patient interactions within a multi-agent framework. For instance, DoctorAgent-RL Feng et al. (2025b) uses such a setup where a "doctor agent" and "patient agent" (termed the "assistant" and "user", respectively, in our work) interact, with the former learning an optimal questioning strategy guided by a comprehensive evaluator. To better guide this learning process, ProMed Ding et al. (2025) introduces a process reward based on Shapley Information Gain, which uses cooperative game theory to formally quantify the clinical utility of each question, enabling more targeted policy optimization. In contrast, Savage Conversation Forests (SCF) Savage (2025) concentrates on the training architecture, which employs a branched conversation structure to help the model learn how early conversational choices impact downstream outcomes by exploring multiple dialogue paths simultaneously. Despite these excellent works, a

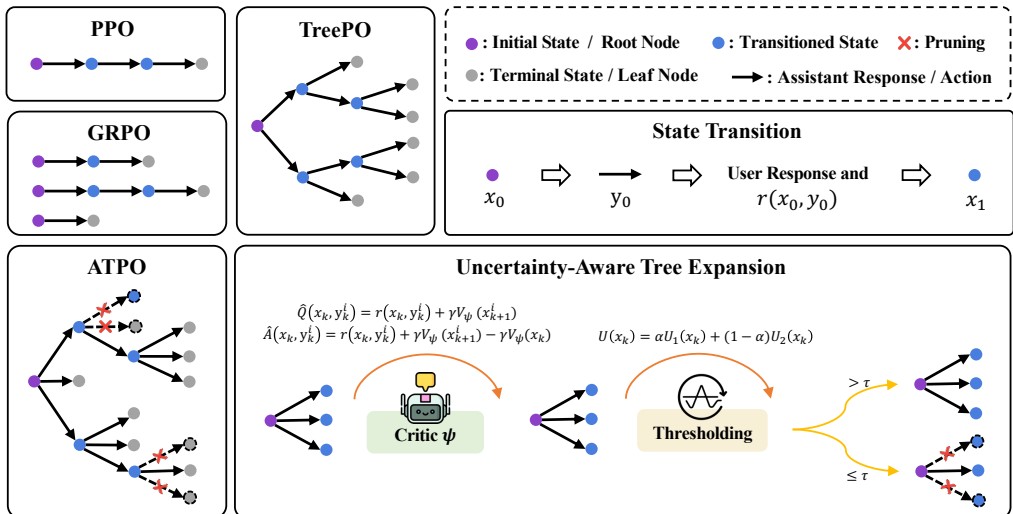

Figure 1: Overview of ATPO algorithm. ATPO generates training data via an adaptive tree search. At each expansion step, it generates candidate child nodes and computes a composite uncertainty score based on their Bellman error $U_1$ and Q-value variance $U_2$. Nodes with high uncertainty $U$ are fully expanded, while low-uncertainty nodes are pruned by randomly selecting a single child for the subsequent rollout. The collected trajectories are then used for policy and critic updates.

robust reinforcement learning algorithm capable of training a model to handle the complexities of real-world medical dialogues remains to be fully explored.

## 2.2 Tree-based Reinforcement Learning

To enhance the reasoning capabilities of LLMs, recent reinforcement learning approaches have begun to integrate tree-based search, particularly for single-turn tasks. These methods primarily focus on three distinct goals. First, some aim to improve value estimation; for example, VinePPO Kazemnejad et al. (2024) uses auxiliary "vine" rollouts to compute more accurate Monte Carlo values, mitigating the impact of inaccurate value estimation. Second, another line of work leverages model uncertainty to guide exploration, typically by expanding the search tree at tokens with high entropy (Hou et al., 2025; Dong et al., 2025). Third, others refine the search structure itself for better credit assignment or efficiency, such as SPO Guo et al. (2025b), which defines "segments" based on low-probability tokens, or TreePO Li et al. (2025b), which uses a fixed N-ary tree for computational gains. Despite their innovations, these methods are fundamentally limited by their single-turn, token-level focus. Their operational units (tokens, segments) do not naturally translate to the macro-level decisions required in multi-turn dialogue. Moreover, their uncertainty metrics are either indirect proxies (e.g., token entropy) or absent entirely in methods with fixed structures. Our approach directly addresses these limitations. We introduce an uncertainty measure based on the variance of turn-level Q-values, which quantifies the uncertainty over future rewards for different macro-actions (i.e., conversational turns). This makes our method inherently better suited for the long-horizon planning challenges of multi-turn interactions.

## 3 Methods

### 3.1 Multi-Turn Dialogues as a Hierarchical MDP

Similar to ArCHer Zhou et al. (2024), we model multi-turn dialogues as a Hierarchical Markov Decision Process (H-MDP), which comprises a high-level MDP with a nested low-level MDP. For the high-level MDP, a macro-action is defined as the full token sequence of the assistant's response in a single turn. A micro-action in the low-level MDP is a single token, such that generating a sequence of micro-actions in the low-level MDP leads to a single macro-action in the high-level MDP. Formally, each state $x_k$ in the high-level MDP consists of the interaction history between the

user and the assistant prior to the $k$-th turn, together with the user's query $q_k$ at the $k$-th turn. The macro-action $y_k$ is defined as a variable-length token sequence representing the assistant's response to $q_k$. The low-level MDP models the process of generating a macro-action, where each low-level action $y_{k,t}$ corresponds to an individual token (i.e., the $t$-th token in the $k$-th macro-action). A low-level state $x_k, t$ is defined as the concatenation of the high-level state $x_k$ and the partial sequence $y_{k,<t}$, namely the tokens that have already been generated in the current turn up to (but excluding) the $t$-th token.

## 3.2 DIALOGUE EXPLORATION VIA TREE EXPANSION

In H-MDP optimization, accurate state-value estimation is crucial for effective policy improvement. Existing methods either rely on pure Monte Carlo estimates from full trajectories (e.g., GRPO Shao et al. (2024)), which can be high-variance and lead to unstable training, or solely on a learned critic (e.g., PPO Schulman et al. (2017)), which may suffer from approximation errors. To strike a balance between accuracy and efficiency, we propose an exploration strategy based on incremental tree expansion. This approach efficiently reuses the computation of shared dialogue prefixes (a detailed analysis of the computational savings is provided in Appendix A.7) and adaptively allocates the sampling budget to the most promising or uncertain parts of the dialogue space, rather than repeatedly re-exploring from the root.

In this framework, the multi-turn dialogue process is viewed as the expansion of a search tree. The initial user query forms the root node, which is at depth $0$ and corresponds to state $x_0$. From there, the tree expands turn-by-turn. At any non-terminal node corresponding to state $x_k$, the assistant produces a macro-action, which can be either a clarifying question or a definitive answer. In the case of a clarifying question, the user's reply completes the state transition to $x_{k+1}$, resulting in the creation of a new node at the next depth of the tree. In the case of a definitive answer, the dialogue terminates along that branch and the node becomes a terminal leaf; for consistency, the subsequent user feedback in this scenario is considered empty. Thus, the depth of a node corresponds to the number of turns elapsed in the dialogue, with the root at depth $0$, and the nodes at depth $k$ representing all possible dialogue states after $k$ turns of interaction.

This entire procedure is guided by an uncertainty-aware principle. The core idea is to prioritize the expansion of nodes exhibiting high uncertainty, as they provide more diverse and informative samples for driving effective policy updates and improving critic accuracy. In the multi-turn dialogue setting, this uncertainty can be broadly categorized into two types: **Epistemic uncertainty**, arising from the inherent cognitive limitations of the model and manifests as uncertainty in its action. **Aleatoric uncertainty**, stemming from the inherent randomness in the environment, such as the variability in user responses.

## 3.3 UNCERTAINTY-AWARE TREE EXPANSION MECHANISM

To operationalize our tree expansion strategy, we first quantify the uncertainty of each frontier node. Consider a specific node at depth $k$, representing the dialogue state $x_k$. We sample a set of $N$ candidate macro-actions $\{y_k^i\}_{i=1}^N$ from the policy $\pi_\theta(\cdot|x_k)$. For each candidate action $y_k^i$, we define its estimated action-value $\hat{Q}(x_k, y_k^i)$ using a one-step lookahead:

$$\hat{Q}(x_k, y_k^i) = r(x_k, y_k^i) + \gamma V_\psi(x_{k+1}^i), \tag{1}$$

where $x_{k+1}^i$ is the state resulting from action $y_k^i$, and the next-state value $V_\psi$ is given by the critic model. Based on this, the uncertainty is calculated as:

$$U_1(x_k) = \left| V_\psi(x_k) - \frac{1}{N} \sum_{i=1}^N \hat{Q}(x_k, y_k^i) \right|, \tag{2}$$

$$U_2(x_k) = \mathrm{Var}_{i \in [N]} \left[ \hat{Q}(x_k, y_k^i) \right] = \frac{1}{N} \sum_{i=1}^N \left( \hat{Q}(x_k, y_k^i) - \frac{1}{N} \sum_{j=1}^N \hat{Q}(x_k, y_k^j) \right)^2, \tag{3}$$

$$U(x_k) = \alpha \, U_1(x_k) + (1 - \alpha) \, U_2(x_k). \tag{4}$$

The first term, $U_1(x_k)$, measures the Bellman error between the critic's current state-value estimate $V_\psi(x_k)$ and the empirical one-step lookahead value averaged over all candidates. It serves as a

proxy for **aleatoric uncertainty**; a large error indicates an inaccurate value estimate for the current state. Note that this error metric is used solely as a heuristic to guide the tree expansion process, rather than as a target for bootstrapping updates of the critic model. The second term, $U_2(x_k)$, quantifies the variance of the action-value estimates. This term captures a blend of **both epistemic and aleatoric uncertainty**. High variance can arise from two distinct sources: the policy's own indecision leading it to explore a diverse set of candidate actions (epistemic), or the environment's inherent randomness where different state transitions yield widely varying Q-values (aleatoric). The notation $\text{Var}(\cdot)$ represents the variance across the $N$ candidate actions. This raw value is subsequently normalized using Z-score scaling based on historical samples to create a stable, comparable metric. The hyperparameter $\alpha \in [0, 1]$ balances the contributions of these two uncertainty sources.

With the uncertainty metric $U(x_k)$ defined, the expansion process is guided by a threshold-based rule. When the expansion process reaches a node at depth $k$, representing one of the possible dialogue states after $k$ turns, we compare its calculated uncertainty against a predefined threshold $\tau$: **If** $U(x_k) > \tau$, the node is considered highly uncertain and all $N$ branches are retained. **Conversely, if** $U(x_k) \leq \tau$, the node is deemed sufficiently understood. To conserve computational resources, we primarily prune the search by randomly selecting only one of the $N$ candidate branches. However, to maintain a baseline level of sampling diversity, we bypass this pruning with a small probability (e.g., $10\%$) and instead expand all $N$ branches.

The uncertainty-driven expansion continues along the retained branches until all dialogues terminate or the total number of leaf nodes (i.e., the tree width) reaches a predefined budget. Once this budget is met, no further nodes are expanded, and all current leaf nodes proceed to the rollout phase until termination.

## 3.4 VALUE TRACEBACK, TREE DECOMPOSITION AND MODEL OPTIMIZATION

Following the completion of the tree expansion process, we compute the state-values and advantages for all nodes within the generated tree via a recursive backward pass starting from the leaf nodes. First, we calculate the target value $\hat{V}(x_k)$ for each state $x_k$, which serves as a robust empirical estimate of the Monte Carlo return from that state and can be used as the value target for subsequent critic updates:

$$\hat{V}(x_k) = \begin{cases} r(x_k, y_k), & \text{if leaf node,} \\ \dfrac{1}{B(x_k)} \displaystyle\sum_{i=1}^{B(x_k)} \left[ r(x_k, y_k^i) + \gamma \hat{V}(x_{k+1}^i) \right], & \text{otherwise.} \end{cases} \tag{5}$$

If a state is terminal, its target value equals the immediate reward; otherwise, it is the average one-step TD target over all child branches. Here, $B(x_k)$ denotes the number of child nodes, which according to the expansion rule is either $N$ for fully expanded nodes or 1 for pruned nodes, and the superscript $i$ indexes the $i$-th branch among these children.

With the target values established, we compute the estimated advantage for each macro-action $y_k^i$ using the standard one-step temporal-difference formulation:

$$\hat{A}(x_k, y_k^i) = r(x_k, y_k^i) + \gamma\, V_\psi(x_{k+1}^i) - V_\psi(x_k). \tag{6}$$

The reason we use the critic's value estimates instead of the target values $\hat{V}(\cdot)$ is that, when only one branch is retained, $\hat{V}(x_k)$ equals its one-step return, resulting in a zero advantage; whereas the critic's estimates preserve non-zero learning signals.

Once values and advantages have been computed for all nodes, the expanded tree is decomposed into a set of independent trajectories for model optimization. Each unique path from the root to a leaf node constitutes one trajectory, so a tree with $M$ leaf nodes directly yields $M$ such trajectories. The number of turns in the $j$-th trajectory, denoted by $K^{(j)}$, can vary across trajectories because different dialogues may terminate after different numbers of turns. The policy $\pi_\theta$ is then updated by

maximizing the following PPO-style objective, aggregated over all tokens from all trajectories:

$$J(\theta) = \mathbb{E}_{x_0 \sim \mathcal{D}, \{y_k \sim \pi_\theta(x_k), x_{k+1} \sim P(\cdot|x_k, y_k)\}_{k=0}^{K-1}} \left[ \frac{1}{M} \sum_{j=1}^{M} \frac{1}{K^{(j)}} \sum_{k=1}^{K^{(j)}} \frac{1}{C(x_k^{(j)}) \, L(y_k^{(j)})} \right. \tag{7}$$

$$\left. \sum_{t=1}^{L(y_k^{(j)})} \min\left(\rho_{k,t}^{(j)} \hat{A}(x_k, y_{k,t}^{(j)}), \mathrm{clip}\left(\rho_{k,t}^{(j)}, 1-\epsilon, 1+\epsilon\right) \hat{A}(x_k, y_{k,t}^{(j)})\right) - \beta D_{\mathrm{KL}}(\pi_\theta || \pi_{\mathrm{ref}}) \right],$$

$$\text{where } \rho_{k,t}^{(j)} = \frac{\pi_\theta\left(y_{k,t}^{(j)} \,\Big|\, x_k^{(j)}, y_{k,<t}^{(j)}\right)}{\pi_{\mathrm{ref}}\left(y_{k,t}^{(j)} \,\Big|\, x_k^{(j)}, y_{k,<t}^{(j)}\right)}, \text{ and } \hat{A}(x_k, y_{k,t}^{(j)}) = \hat{A}(x_k, y_k^{(j)}) \text{ for } 1 \le t \le L(y_k^{(j)}).$$

In this objective, the expectation $\mathbb{E}(\cdot)$ is taken over initial user queries $x_0$ sampled from the dataset $\mathcal{D}$, along with subsequent states and actions. The term $\rho_{k,t}^{(j)}$ denotes the probability ratio between the current policy $\pi_\theta$ and the reference policy $\pi_{\mathrm{ref}}$ for generating the $t$-th token in the $k$-th turn of the $j$-th trajectory. Consistent with our hierarchical MDP formulation, the advantage $\hat{A}(x_k, y_{k,t}^{(j)})$ for each token is set equal to the macro-action's advantage $\hat{A}(x_k, y_k^{(j)})$, thereby uniformly distributing the turn-level credit across the tokens generated in that turn. The update is normalized by two factors: $C(x_k^{(j)})$, the visit count of the state across all trajectories, which prevents over-optimizing on frequently visited nodes; and $L(y_k^{(j)})$, the length of the response, which ensures the turn-level advantage is properly scaled for each token. The $\min(\cdot)$ and $\mathrm{clip}(\cdot)$ constrain the policy update to stabilize training.

Finally, the critic model, parameterized by $\psi$, is trained to predict the target state values. The critic consists of the LLM backbone and a linear value head. Its value estimate $V_\psi(x_k)$ is obtained as the average of the predictions over the final $h$ special tokens. The critic is trained by minimizing the following mean squared error loss:

$$\mathcal{L}(\psi) = \frac{1}{M} \sum_{j=1}^{M} \frac{1}{K^{(j)} \cdot h} \sum_{k=1}^{K^{(j)}} \sum_{t=L(x_k^{(j)})-h+1}^{L(x_k^{(j)})} \frac{1}{2} \left[ V_\psi(x_{k,t}^{(j)}) - \hat{V}(x_k^{(j)}) \right]^2, \tag{8}$$

where $V_\psi(x_{k,t}^{(j)})$ is the value predicted at the $t$-th token position of the input, with $L(x_k^{(j)})$ denoting the total number of tokens of state $x_k^{(j)}$. This loss trains the critic to make the predictions at the final $h$ token positions for state $x_k^{(j)}$ match the ground-truth target value $\hat{V}(x_k)$.

## 4 EXPERIMENTS

### 4.1 EXPERIMENT SETUP

**Environment and Task Description.** We establish a multi-turn clinical case reasoning environment with two LLM-based agents: a User Simulator and an Assistant Agent. (1) **User Simulator**: Implemented using Qwen3-8B, it is instructed to answer the Assistant's questions strictly based on a given set of atomic facts (an example is provided in Appendix A.9), refusing to respond to any out-of-scope queries. The prompt for user simulator is provided Appendix A.8. To ensure reliability, we continuously monitor its behavior during training and verify it with GPT-4o, achieving 100% accuracy in following instructions and rejecting irrelevant queries, with a hallucination rate of only 1.2%, demonstrating high fidelity. This verification was performed using the user simulator quality monitoring prompt (see Appendix A.8). (2) **Assistant Agent**: The Assistant Agent is tasked with resolving a clinical case question by selecting the correct answer from a list of options. It begins with an initial context and can iteratively query the User Simulator for additional information. The process terminates when the agent either commits to a final answer or the pre-defined turn limit is reached. An interaction example between the Assistant Agent and the User Simulator is provided in Appendix A.1.

**Baselines.** We compare our ATPO with several baselines to evaluate its effectiveness:

1) **Zero-shot Prompting**: To benchmark performance, we evaluate several base models (Qwen3-1.7B, Qwen3-4B, and Qwen3-8B) under two distinct settings. The first is a Direct single-turn setting, where the agent must respond using only the initial context. The second is an interactive MEDIQ Li et al. (2024) setting, which allows the agent to interact with the User Simulator for up to 8 turns in total, including the final answering turn. Prompts for both settings are provided in Appendix A.8.

2) **SFT**: To establish a stronger baseline, we fine-tune the Qwen3 instruct models to encourage multi-turn information-seeking behavior instead of directly producing an answer in the first turn. Using the MEDIQ validation dataset as the source of atomic facts, questions, and answer options, we employ the expert model Gemini-2.5-Pro in a self-play setup, where it role-plays both user and assistant to generate $1,269$ multi-turn dialogues (the prompt is provided in code repository as it is too long to display in the paper). We ensure no information leakage during this process and retain only dialogues with a correct final answer. The resulting dataset is used to train the models with supervised fine-tuning and dynamic fine-tuning (DFT Wu et al. (2025b)).

3) **SFT+RL**: We compare our algorithm with standard RL post-training methods under the same environment, using rewards solely based on the correctness of the final answer. *Critic-based Methods*: PPO (MDP) treats text generation as a standard MDP, assigning a unique value to each token. PPO (H-MDP) adopts a hierarchical formulation, estimating a single value per turn and propagating the corresponding advantage to all tokens in that turn. *Critic-free Methods*: We implement GRPO, which assigns a single advantage to an entire trajectory and shares it across all tokens. We also adopt TreePO (Guo et al., 2025b; Li et al., 2025b), where we construct a binary search tree in which each non-terminal node expands into two child nodes, constrained only by a maximum depth corresponding to the dialogue turn limit, without any pruning. After the search completes, we perform a backward pass to compute the aggregate return for each node as its target value $\hat{V}(x_k)$. The advantage is calculated as $\hat{A}(x_k, y_k) = r(x_k, y_k) + \gamma \hat{V}(x_{k+1}) - \hat{V}(x_k)$, and is assigned to all tokens generated in that sequence for policy optimization.

**Implementation Details.** For RL experiments, we use the SFT-trained Qwen3-1.7B, Qwen3-4B, and Qwen3-8B models as the initial policy for the Assistant Agent. The reward function is based solely on final-answer correctness: $+3$ for a correct answer, $0$ for an incorrect answer, and $-1$ for an invalid format. The training data contains $14,256$ samples, with $66\%$ ($9,400$) drawn from the MEDIQ training dataset and $34\%$ ($4,856$) constructed from the MedMCQA training dataset.

Policy learning rate $1 \times 10^{-6}$, critic learning rate $1 \times 10^{-5}$, KL penalty weight $\beta = 0.01$, and discount factor $\gamma = 1$. The critic is initialized from the actor's weights and warmed up for 5 steps. Method-specific settings include a group size of 32 for GRPO, and for ATPO, an expansion size $N = 4$ with a total expansion budget of 128. In ATPO ($U_1$), the uncertainty threshold is $\tau = 0.5$; in ATPO ($U_1 + U_2$), we set $\alpha = 0.3$ and $\tau = 1.5$.

Our TreePO and ATPO implementations build upon the VeRL (Sheng et al., 2025) Agentic RL framework, integrating tree search, reward computation, and advantage estimation into a single concurrent phase. This design eliminates the need for a multi-stage pipeline by producing ready-to-train trajectories directly from the search process. In ATPO, throughput is further improved by executing the assistant model's answer generation, its interaction handling with the user model, and the value estimation by the critic model in a fully asynchronous manner within the sampling phase. Together with efficient prefix sharing and the vLLM KV cache, this design achieves decoding speeds of up to 2,500 tokens/sec/GPU on a 1.7B model with TreePO. Both our implementation and the associated datasets are available at https://github.com/Quark-Medical/ATPO.

## 4.2 RESULTS

**Evaluation Setup.** We conduct evaluations on three Qwen3 models Yang et al. (2025) of different sizes (Qwen3-1.7B, Qwen3-4B, and Qwen3-8B), along with GPT-4o and Gemini-2.5-Pro as a strong baseline to assess the effectiveness of our method. Experiments are performed on three multi-turn medical dialogue datasets adapted from public multiple-choice question datasets: MedQA, obtained directly from the MEDIQ Li et al. (2024) test set; MedMCQA, constructed from its original training data Pal et al. (2022); and MedicalExam, sourced directly from AIE Liao et al. (2024). Each sample is reformulated into a set of atomic facts, a concise initial context, an atomic question that excludes factual details, and several answer options with exactly one correct choice (details in Appendix A.2). The primary evaluation metric is *final-answer accuracy*, defined as the percentage of test cases where

Table 1: Performance comparison (%) on **MedicalExam**, **MedQA**, and **MedMCQA**. **Bold** indicates the best performance, underlined the second-best (excluding GPT-4o and Gemini-2.5-Pro).

| Model | Method Type | Method Name | MedicalExam | MedQA | MedMCQA |
|---|---|---|---|---|---|
| **Qwen3-1.7B** | Prompt | Direct | $35.07 \pm 1.12$ | $34.05 \pm 0.38$ | $32.54 \pm 0.49$ |
| | | MEDIQ | $34.00 \pm 2.26$ | $34.20 \pm 0.75$ | $32.35 \pm 1.73$ |
| | SFT | DFT | $29.07 \pm 1.46$ | $28.38 \pm 0.80$ | $21.08 \pm 0.90$ |
| | | SFT | $32.27 \pm 4.77$ | $33.42 \pm 0.95$ | $28.10 \pm 2.32$ |
| | SFT+RL | PPO (MDP) | $39.33 \pm 4.01$ | $38.64 \pm 1.17$ | $35.37 \pm 0.80$ |
| | | PPO (H-MDP) | $39.33 \pm 2.79$ | $39.08 \pm 1.85$ | $34.89 \pm 1.00$ |
| | | GRPO | $42.93 \pm 1.80$ | $41.17 \pm 0.64$ | $36.57 \pm 3.26$ |
| | | TreePO | $43.33 \pm 1.56$ | $42.05 \pm 1.03$ | $38.47 \pm 2.00$ |
| | | ATPO ($U_1$) | $\mathbf{45.73 \pm 1.53}$ | $42.54 \pm 0.39$ | $38.66 \pm 0.66$ |
| | | ATPO ($U_1 + U_2$) | $43.20 \pm 1.85$ | $\mathbf{42.87 \pm 0.77}$ | $\mathbf{39.93 \pm 1.05}$ |
| **Qwen3-4B** | Prompt | Direct | $48.13 \pm 0.87$ | $44.94 \pm 0.35$ | $41.53 \pm 0.39$ |
| | | MEDIQ | $45.87 \pm 1.20$ | $40.11 \pm 0.60$ | $31.64 \pm 1.41$ |
| | SFT | DFT | $43.07 \pm 1.61$ | $41.72 \pm 1.27$ | $33.28 \pm 1.68$ |
| | | SFT | $48.93 \pm 2.14$ | $47.15 \pm 1.01$ | $39.18 \pm 1.22$ |
| | SFT+RL | PPO (MDP) | $50.13 \pm 2.80$ | $50.60 \pm 0.90$ | $42.50 \pm 0.84$ |
| | | PPO (H-MDP) | $52.40 \pm 2.24$ | $48.58 \pm 1.48$ | $43.32 \pm 2.22$ |
| | | GRPO | $53.87 \pm 2.08$ | $51.17 \pm 1.08$ | $43.84 \pm 0.78$ |
| | | TreePO | $56.13 \pm 0.99$ | $53.74 \pm 0.56$ | $45.22 \pm 0.65$ |
| | | ATPO ($U_1$) | $56.80 \pm 1.28$ | $\underline{53.15 \pm 0.55}$ | $\mathbf{46.23 \pm 1.25}$ |
| | | ATPO ($U_1 + U_2$) | $\mathbf{59.73 \pm 2.61}$ | $\mathbf{55.47 \pm 0.99}$ | $\underline{45.93 \pm 1.13}$ |
| **Qwen3-8B** | Prompt | Direct | $52.40 \pm 0.37$ | $45.22 \pm 0.34$ | $46.16 \pm 1.04$ |
| | | MEDIQ | $51.87 \pm 3.69$ | $46.03 \pm 0.75$ | $41.60 \pm 0.91$ |
| | SFT | DFT | $51.86 \pm 3.63$ | $48.80 \pm 1.30$ | $42.20 \pm 0.83$ |
| | | SFT | $55.87 \pm 0.30$ | $53.75 \pm 1.18$ | $46.87 \pm 1.74$ |
| | SFT+RL | PPO (MDP) | $59.20 \pm 3.84$ | $57.38 \pm 0.84$ | $50.00 \pm 0.81$ |
| | | PPO (H-MDP) | $59.07 \pm 3.15$ | $57.81 \pm 1.29$ | $51.98 \pm 0.67$ |
| | | GRPO | $60.93 \pm 1.86$ | $57.92 \pm 0.68$ | $51.12 \pm 1.29$ |
| | | TreePO | $65.33 \pm 3.09$ | $61.81 \pm 0.90$ | $\mathbf{54.74 \pm 1.99}$ |
| | | ATPO ($U_1$) | $65.52 \pm 3.12$ | $62.57 \pm 0.41$ | $53.22 \pm 1.30$ |
| | | ATPO ($U_1 + U_2$) | $\mathbf{65.87 \pm 3.72}$ | $\mathbf{64.07 \pm 0.43}$ | $53.66 \pm 1.52$ |
| **GPT-4o** | Prompt | MEDIQ | $64.00 \pm 3.53$ | $63.15 \pm 0.82$ | $53.03 \pm 0.89$ |
| **Gemini-2.5-Pro** | Prompt | MEDIQ | $74.33 \pm 2.53$ | $68.69 \pm 0.61$ | $63.31 \pm 1.37$ |

the Assistant Agent's chosen option matches the ground-truth answer. For statistical robustness, we report the mean and standard deviation of five independent runs.

**Main Findings.** From Table 1, we observe that in the zero-shot setting, the MEDIQ prompting strategy performs worse than the Direct single-turn prompt, consistent with the finding in MEDIQ Li et al. (2024) that prompting LLMs to ask questions can reduce accuracy. Supervised fine-tuning brings only modest gains in final-answer accuracy while being crucial for enabling multi-turn information seeking and providing a solid foundation for subsequent reinforcement learning. We further explored the limits of SFT by distilling knowledge from much larger models like GPT-4o and Gemini-2.5-Pro. As detailed in Appendix A.3, this distillation approach also failed to yield significant improvements (Table 2), reinforcing the notion that simply imitating expert trajectories is insufficient for generalization and that a goal-driven reinforcement learning approach is necessary.

Our proposed ATPO achieves the highest accuracy in most settings, even surpassing GPT-4o at the 8B scale (e.g., exceeding GPT-4o on MedQA by $0.92\%$). This demonstrates the strong effectiveness of the method. Further, the results show that both uncertainty metrics are valuable and complementary: ATPO ($U_1 + U_2$) generally outperforms ATPO ($U_1$), which in turn achieves higher accuracy than TreePO. Combining $U_1$ and $U_2$ yields the best results, with absolute gains over TreePO on MedQA of $0.82\%$, $1.73\%$, and $2.26\%$ for the 1.7B, 4B, and 8B models, respectively. This performance improvement is partly driven by the enhanced quality of the dialogues. As shown in Appendix A.4, the proportion of effective questions asked by the assistant steadily increases during ATPO training (Figure 4), allowing it to gather crucial information more efficiently and solve tasks in fewer turns.

ATPO also exhibits markedly higher sample efficiency, as shown in Figure 2 (a). For instance, on MedQA with Qwen3-4B, ATPO ($U_1 + U_2$) achieves approximately $52.7\%$ accuracy while using only $55\%$ of the training turns required by TreePO (A detailed analysis of the computational costs of

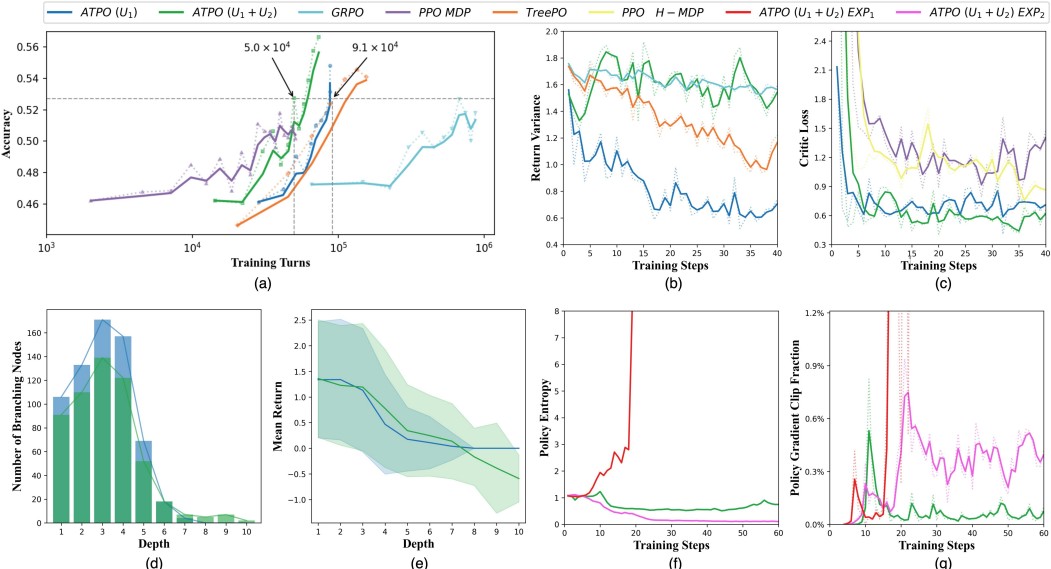

Figure 2: Analysis of the ATPO algorithm on Qwen3-4B. (a) Training efficiency and performance comparison of various algorithms, plotting accuracy against the number of generated turns. (b), (c) Return variance and critic loss for ATPO and baseline methods. (d), (e) Distribution of branching nodes and returns by depth for samples from ATPO at a representative training step. (f), (g) Stability analysis of ATPO with and without visit-count-based down-weighting.

various algorithms is provided in A.5). This advantage stems from ATPO's adaptive rollout budget allocation mechanism. Moreover, unlike TreePO's fixed branching, which causes an exponential growth of nodes concentrated in early turns, ATPO's uncertainty-based pruning enables deeper and more balanced exploration across all dialogue depths (as shown in Figures 2 (d), (e)), making it more suitable for long-horizon tasks. Additionally, hierarchical modeling proves beneficial in multi-turn dialogue: PPO (H-MDP) slightly but consistently surpasses PPO (MDP), scoring higher in 5 out of 9 evaluation settings. Among critic-free methods, the tree-based approach demonstrates clear superiority, with TreePO substantially outperforming GRPO. This indicates that, for complex multi-turn tasks, structuring credit assignment via a search tree is more effective than relying on a single trajectory-level advantage.

## 4.3 ABLATION AND ANALYSIS

We conducted several ablation experiments on ATPO, which provide the following key insights.

The dual uncertainty metrics enhance both sampling diversity and critic optimization. Figure 2 (b) shows that guiding node expansion with $U_1 + U_2$ produces a high variance of sample returns, comparable to GRPO and markedly higher than TreePO, while using only $U_1$ reduces diversity. At the same time, Figure 2 (c) indicates that the critic's value loss under $U_1 + U_2$ is substantially lower than PPO (both MDP and H-MDP), with $U_1$ alone ranking second. These results highlight that uncertainty-aware tree search benefits value function learning. This advantage partly stems from intelligent budget allocation (Figures 2 (d), (e)): $U_1$ alone drives aggressive early exploration, concentrating expansions at shallow depths (3–4) and causing steep local drops in node values, whereas $U_1 + U_2$ achieves deeper coverage and maintains a more uniform value variance, enabling a more globally effective search.

We also find that down-weighting policy updates by node visit count is crucial for training stability. An ablation study compares the default ATPO (policy advantage down-weighted) with two variants: $EXP_1$ (no policy down-weighting) and $EXP_2$ (down-weighting both policy advantage and value loss). $EXP_1$ results in uncontrolled entropy growth and excessive policy clipping (Figures 2 (f), (g)), since ignoring visit counts leads to disproportionate policy updates on frequently visited nodes, causing the policy to diverge rapidly from its reference state. In contrast, $EXP_2$ induces rapid en-

tropy collapse. High-visit nodes provide the most reliable value estimates due to abundant samples, and underweighting the value loss on these nodes destabilizes the critic, increasing value variance in early layers over time. As a result, the policy learns to distrust its flawed value function, abandons multi-turn exploration, and regresses to suboptimal single-turn strategies.

Moreover, to assess the generalization of our method, we evaluated the assistant model trained with ATPO using an unseen user simulator. Specifically, for evaluation, we replaced the Qwen3-8B simulator used during training with the Llama-3.3-70B-Instruct model. As detailed in Appendix A.6, the assistant maintains similar performance across all test sets (Table A.6), providing strong evidence that our method does not overfit to the conversational patterns of a specific simulator and possesses robust generalization capability.

## 5 CONCLUSIONS AND FUTURE WORK

In this paper, we present ATPO, a novel adaptive tree search method that intelligently guides exploration in multi-turn dialogues through state-uncertainty evaluation. By selectively expanding nodes that maximize sampling diversity and benefit critic optimization, ATPO achieves superior performance with markedly fewer exploration steps, surpassing strong RL baselines such as TreePO and GRPO across several clinical dialogue benchmarks, and even outperforming GPT-4o on MedQA with the Qwen3-8B model. Beyond its success in multi-turn medical dialogue, ATPO can also be applied to a wide range of scenarios, including multi-turn open-ended dialogue and tool use.

Future work could pursue several promising directions. First, replacing the current fixed-threshold expansion mechanism with a learnable, soft control policy may reduce hyperparameter tuning effort and enable the expansion strategy to adapt dynamically as the policy evolves. This idea could be further extended to adaptively determine the number of nodes to expand based on uncertainty metrics, rather than relying on random selection. Second, refining credit assignment within the Hierarchical MDP (H-MDP) framework could yield additional gains. A more sophisticated approach to distributing high-level advantages to low-level token actions, moving beyond uniform cloning, may allow for more precise policy optimization and improved performance.

## ACKNOWLEDGMENTS

This work was supported by Qwen Applications Business Group of Alibaba Group through Alibaba Research Intern Program.

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

## A APPENDIX

### A.1 INTERACTION PROCESS BETWEEN ASSISTANT AGENT AND USER SIMULATOR

Figure 3 presents an example of the interaction flow in our multi-turn clinical reasoning environment. The process starts from an incomplete initial user query representing a clinical case. The Assistant Agent receives this query along with a set of answer options. The initial information is intentionally limited; attempting to answer directly at this stage often leads to incorrect results. Therefore, the Assistant Agent must actively ask targeted questions to the User Simulator in order to obtain the missing key facts. The User Simulator responds strictly within the scope of predefined atomic facts, refusing to provide any information beyond this scope. This iterative exchange continues until the Assistant Agent commits to a final answer or the predefined turn limit is reached.

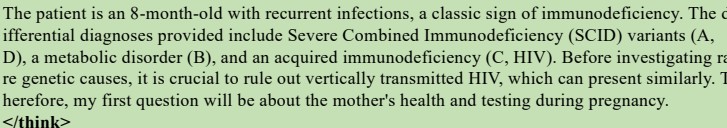

An 8-month-old boy is brought to the office by his mother for recurrent infections.
**Problem:** What is the most likely cause of this patient's condition?
**Options:** {'A': 'Adenosine deaminase deficiency', 'B': 'Hypoxanthine-guanine phosphoribosyl transferase (HGPRT) deficiency', 'C': 'HIV infection', 'D': 'Purine nucleoside phosphorylase deficiency'}

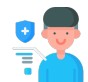

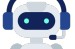

**<think>**
The patient is an 8-month-old with recurrent infections, a classic sign of immunodeficiency. The differential diagnoses provided include Severe Combined Immunodeficiency (SCID) variants (A, D), a metabolic disorder (B), and an acquired immunodeficiency (C, HIV). Before investigating rare genetic causes, it is crucial to rule out vertically transmitted HIV, which can present similarly. Therefore, my first question will be about the mother's health and testing during pregnancy.
**</think>**
**Question:** Was the mother tested for any infectious diseases, such as HIV, during her pregnancy?

The mother denies any history of infection during her pregnancy and was tested negative for HIV.

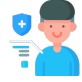

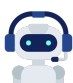

**<think>**
The mother's negative HIV status makes congenital HIV (Option C) highly unlikely. The focus now shifts to primary immunodeficiencies like Adenosine deaminase deficiency (A) and Purine nucleoside phosphorylase deficiency (D), both of which can cause SCID. A key radiological finding in many forms of SCID is the absence of a thymic shadow due to a lack of T-cells. Asking about imaging is the most direct way to investigate this possibility and differentiate from other causes of infection.
**</think>**
**Question:** Has the patient undergone any imaging tests, such as a chest X-ray?

Yes, a chest X-ray reveals an absent thymic shadow and fails to show any lung pathology.

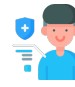

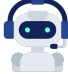

**<think>**
The dialogue has established two critical facts: 1) The mother is HIV negative, ruling out congenital HIV infection (Option C). 2) A chest X-ray revealed an absent thymic shadow. The combination of recurrent infections in early infancy and an absent thymic shadow is a classic presentation of Severe Combined Immunodeficiency (SCID). Adenosine deaminase (ADA) deficiency (Option A) is a well-known cause of autosomal recessive SCID and perfectly matches this clinical picture. Hypoxanthine-guanine phosphoribosyl transferase (HGPRT) deficiency (Option B) presents with neurological and behavioral issues, not immunodeficiency. Purine nucleoside phosphorylase deficiency (Option D) is another, rarer cause of immunodeficiency. Based on the hallmark finding of an absent thymus, ADA deficiency is the most probable diagnosis.
**</think>**
**Final Answer:** A

Figure 3: Schematic diagram of the interaction flow between the Assistant Agent and the User Simulator in the multi-turn clinical reasoning environment. The process starts from an incomplete initial user query, after which the Assistant Agent asks targeted questions and the User Simulator responds strictly within the scope of predefined atomic facts, until a final answer is produced or the turn limit is reached.

### A.2 TEST DATASETS

**MedicalExam**: This test set is curated from a collection provided by Liao et al. (2024), comprising five distinct data sources: MedQA, MedMCQA, MMLU, SelfExam, and QMAX. The original data

from MedMCQA and MMLU lacked the atomic facts. To address this, we employed Gemini-2.5-pro to decompose the original problems into our required structure, consisting of an atomic question, atomic facts, and several answer options, using the prompt detailed in Appendix A.8. The final curated set contains 150 samples.

**MEDQA**: This dataset is derived from the medical dialogues test set provided by MEDIQ ( Li et al. (2024)). We preprocessed this data by filtering out all samples where the atomic facts were empty. The final test set contains a total of $1,268$ samples.

**MedMCQA**: This test set was constructed from the official validation set of MedMCQA ( Pal et al. (2022)). We first selected samples where the question description exceeded 150 characters in length. For these selected samples, we then utilized an LLM to synthesize the corresponding atomic facts and question. This process resulted in a final test set of 536 samples.

## A.3 DISTILLATION EXPERIMENT AND ANALYSIS

Since prompting GPT-4o alone yields results close to the best performance across all benchmarks, a natural idea is to leverage high-quality multi-turn dialogue data generated by GPT-4o or Gemini-2.5-Pro for model distillation. Therefore, we used the same self-play prompts to have both GPT-4o and Gemini-2.5-Pro strictly simulate dialogues based on the data from the RL training dataset (14,256 samples). Then, we perform SFT on the Qwen3-8B model separately using the dialogue data generated by the two models (i.e., distilling the correct dialogue outputs from GPT-4o and Gemini-2.5-Pro separately), and test the performance. As shown in Table 2, even with additional data and extended SFT training time, the model performance shows almost no improvement. This further confirms that the primary role of the SFT stage is to teach the model the question–answer format, and it does not require a particularly large dataset to achieve adequate training. However, it should be noted that the model obtained after SFT training (i.e., the starting checkpoint for RL training) can affect the data obtained during RL sampling, thereby influencing the subsequent RL training Ding et al. (2025); Guo et al. (2025a). It is also evident that SFT training alone, even with large-scale data, fails to give the model strong generalization ability, making RL training necessary for improving its generalization capabilities.

Table 2: The performance of Qwen3-8B distilled from the respective multi-turn dialogue outputs of GPT-4o and Gemini-2.5-Pro.

| Model | Dialogue Simulator | MedicalExam | MedQA | MedMCQA |
|-------|--------------------|-------------|-------|---------|
| Qwen3-8B | GPT-4o | $57.33 \pm 2.26$ | $53.25 \pm 0.72$ | $46.04 \pm 1.31$ |
| | Gemini-2.5-Pro | $58.67 \pm 0.08$ | $51.51 \pm 1.33$ | $48.06 \pm 0.94$ |

## A.4 ANALYSIS OF MODEL EFFECTIVE QUESTION RATE WITH ATPO TRAINING PROGRESS

To further investigate the improvement in dialogue quality brought by ATPO training, we plotted the variation curve of the proportion of effective questions (defined as questions that elicit valid responses from the user) across multi-turn dialogues as training progresses. As illustrated in Figure 4, the proportion of effective questions increases steadily throughout the training process, demonstrating that the assistant's queries become more answerable for the user. This allows the assistant to gather more useful information and accomplish tasks in fewer dialogue turns, thereby enhancing the overall user experience. An example illustrating the change in dialogue quality before and after ATPO training for the Qwen3-8B model is provided in Figure 5.

## A.5 COMPARATIVE ANALYSIS OF THE COMPUTATIONAL REQUIREMENTS OF DIFFERENT ALGORITHMS

To assess whether ATPO's performance gains over other algorithms are simply attributable to greater computational resources, we conducted a detailed evaluation of resource usage across methods. Specifically, we measured the time required by each algorithm to achieve the same test-set accuracy, taking the best performance of PPO as the benchmark. The results are presented in Table 3. As shown, ATPO achieves the target accuracy in the shortest time, whereas GRPO requires the longest.

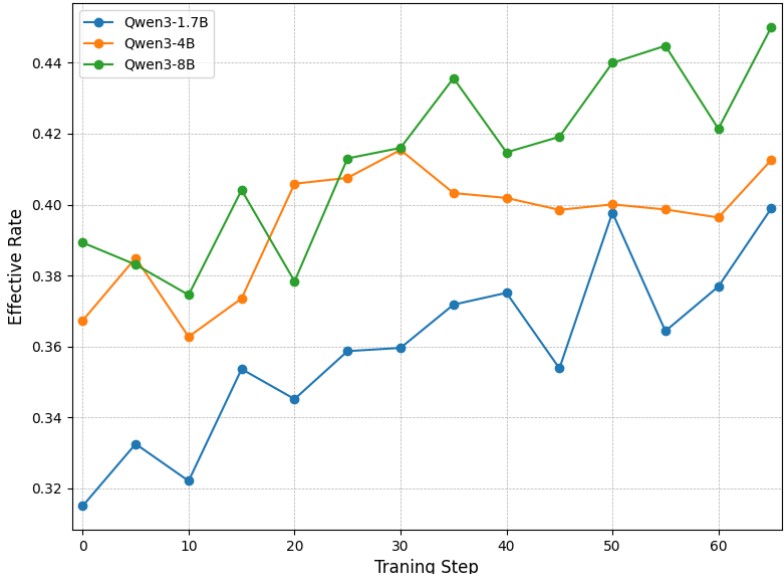

Figure 4: The variation curve of the proportion of effective questions for various scale models across multi-turn dialogues as ATPO training progresses.

Since all algorithms were executed on identical hardware, shorter training time directly translates to reduced computational resource consumption. Therefore, the superior performance of ATPO cannot be attributed to increased computational resources, but rather to its algorithmic effectiveness. Furthermore, in our experiments, we observed that with continued training, PPO's curve had already converged without further improvement, while GRPO and TreePO even experienced entropy explosion and crashed. Additionally, we computed the proportion of total training time spent in the rollout stage for each algorithm, as well as the computational distribution within this stage, as summarized in Table 4. ATPO exhibits the highest rollout-stage proportion, approximately 45% of the total time, in contrast to other methods, which are nearly identical at about 25%. Moreover, more than half of the computation in ATPO's rollout stage is used for value estimation, which is sensible as ATPO frequently estimates node values during rollout to dynamically allocate the rollout budget. However, these costs are meaningful, as it produces high-quality sampling data, thereby accelerating model convergence. This is corroborated by Figure 2(a), where ATPO's convergence curve is steeper than those of other algorithms. In contrast, for other methods, rollout-stage computation is dominated by sequence generation, producing lower-quality sampled data that slows convergence. In summary, while ATPO allocates more time to the rollout stage, its emphasis on value estimation produces higher-quality data, resulting in faster convergence, reduced total training time, and ultimately lower overall computational resource consumption compared to competing algorithms.

| Method | Total Time (hour) | Proportion of Rollout Time |
|---|---|---|
| PPO | 3.02 | 26.16% |
| ATPO ($U_1$+$U_2$) | 2.22 | 45.05% |
| ATPO ($U_1$) | 2.98 | 41.95% |
| TreePO | 3.15 | 24.13% |
| GRPO | 4.86 | 25.72% |

Table 3: The total time required for different algorithms to reach comparable performance, and the proportion of time spent on the rollout phase.

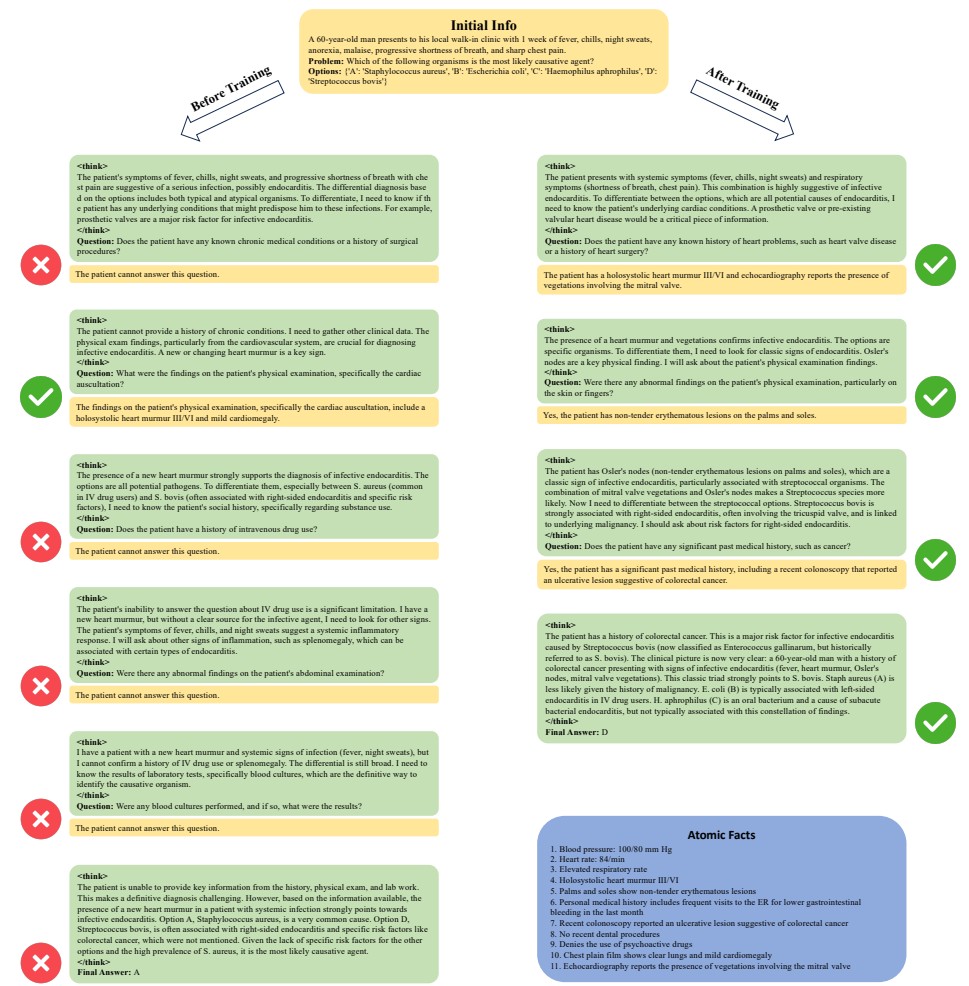

Figure 5: Comparison of model dialogue quality before and after ATPO training.

| Method | Rollout TFLOPs | Proportion of Generation TFLOPs | Proportion of Value Estimation TFLOPs |
|---|---|---|---|
| PPO | $444,915.23$ | $100\%$ | $0\%$ |
| ATPO ($U_1+U_2$) | $639,410.69$ | $37.99\%$ | $62.01\%$ |
| ATPO ($U_1$) | $817,173.44$ | $42.38\%$ | $57.62\%$ |
| TreePO | $916,408.62$ | $100\%$ | $0\%$ |
| GRPO | $4,708,526.96$ | $100\%$ | $0\%$ |

Table 4: Rollout computation required by various algorithms to reach comparable performance, along with the respective proportions allocated to generation and to value estimation.

## A.6 Tested with Different User Simulators

We employed Qwen3-8B as the user simulator in both the training and testing phases. However, relying on a single simulator may lead the assistant model to overfit to its specific tone or response format, potentially limiting generalization to unseen simulators. To assess this risk, we replaced the user simulator at test time with the Llama-3.3-70B-Instruct model, and re-evaluated the performance of Qwen3-1.7B, Qwen3-4B, and Qwen3-8B trained with ATPO ($U_1 + U_2$). As shown in Table 5, the assistant's performance after replacement remains almost unchanged, with no significant differences. These results indicate that the assistant model did not overfit to the tone or response style of the training user simulator and can perform effectively with an unseen simulator, thereby demonstrating strong generalization capability.

| Model | User Simulator | MedicalExam | MedQA | MedMCQA |
|---|---|---|---|---|
| Qwen3-1.7B | Qwen3-8B | $43.20 \pm 1.85$ | $42.87 \pm 0.77$ | $39.93 \pm 1.05$ |
| | Llama-3.3-70B-Instruct | $43.07 \pm 1.53$ | $43.15 \pm 0.60$ | $40.07 \pm 1.39$ |
| Qwen3-4B | Qwen3-8B | $59.73 \pm 2.61$ | $55.47 \pm 0.99$ | $45.93 \pm 1.13$ |
| | Llama-3.3-70B-Instruct | $61.20 \pm 2.76$ | $56.47 \pm 0.94$ | $47.76 \pm 1.06$ |
| Qwen3-8B | Qwen3-8B | $65.87 \pm 3.72$ | $64.07 \pm 0.43$ | $53.66 \pm 1.52$ |
| | Llama-3.3-70B-Instruct | $67.07 \pm 3.35$ | $63.93 \pm 1.01$ | $55.60 \pm 1.21$ |

Table 5: Performance of different models tested with different user simulators.

## A.7 Analysis of the Computational Savings of Tree Search

ATPO performs sampling based on a tree structure, which can reduce the computational cost during the sampling process. Here, we provide a brief theoretical analysis of the potential savings gained through tree expansion, following the computation statistics methodology in VeRL (Sheng et al., 2025).

For an LLM of a given size, let $\phi$ denote the number of parameters in the linear projections and $\theta$ denote the number of parameters in the attention mechanism. For a specific turn in a multi-turn dialogue, denote the input length of the current turn as $x$ and the output length as $y$.

If we sample one data instance at a time, using the KV cache, the *prefill* stage on the input incurs a computational cost:

$$C_{\text{prefill}} = 2\phi x + 4\theta x^2.$$

During decoding, we benefit from cached $K$ and $V$, yielding a computational cost:

$$C_{\text{decode}} = 2\phi y + 2\theta(2x + y + 1).$$

When sampling $N$ instances of equal length, the total computational cost without tree expansion becomes:

$$C_{\text{independent}} = N \cdot \left(2\phi x + 4\theta x^2\right) + N \cdot \left(2\phi y + 2\theta(2x + y + 1)\right).$$

In contrast, with tree expansion, the prefill step is performed only once, followed by decoding $N$ samples. The total cost becomes:

$$C_{\text{tree}} = 1 \cdot \left(2\phi x + 4\theta x^2\right) + N \cdot \left(2\phi y + 2\theta(2x + y + 1)\right).$$

The theoretical computational saving is therefore:

$$\Delta C = (N - 1) \cdot \left(2\phi x + 4\theta x^2\right).$$

Since ATPO is grounded in tree-structured sampling, it inherits this cost reduction property.

It is worth noting that TreePO (Li et al., 2025b) provides a rigorous empirical verification of tree-based sampling efficiency (Section 4.1), showing higher throughput in tokens per second. Compared to TreePO's scenario, where the inference chain is divided into multiple segments for tree-based sampling, our multi-turn setting produces semantically distinct tree nodes, making it naturally more amenable to tree search.

A.8   PROMPTS

**User Prompt**

You are a medical information assistant. Your role is to help doctors by providing information strictly from patient data.

INSTRUCTIONS:

1. Search through the provided atomic facts for information that directly answers the doctor's question
2. If you find relevant atomic facts, provide the answer using ONLY that information
3. Do NOT add any medical analysis, inference, interpretation, or external knowledge
4. Do NOT make assumptions or draw conclusions beyond what is explicitly stated
5. If no atomic fact directly answers the question, respond with exactly this phrase: "The patient cannot answer this question."

Patient atomic facts: {atomic facts}

Doctor's question: {doctor's question}

Your response:

**Direct Method Prompt**

You are an expert medical assistant. Based on the medical case given by user, which includes initial patient information, a question, and several options, select the single best answer. Your response must be only the letter of the chosen option (e.g., A, B, C...), without any additional text, punctuation, or explanation.

Initial information: [initial patient information]

Question: [question]

Options: [options]

Your response:

**MEDIQ System Prompt**

You are a professional medical assistant, possessing outstanding medical diagnostic reasoning and analytical abilities, as well as strong clinical inquiry and patient assessment skills.

Below, the user will provide initial patient information at the beginning of the first turn of conversation, pose a single-choice question (Problem: question description), and corresponding options (Options: option descriptions). Your task is to, based on the question description, the option descriptions, the currently available patient information, and your own knowledge, select the correct option.

Note: The initial patient information provided by the user in the first turn is incomplete. You can ask the user questions to continuously obtain more patient information until you are confident enough to select the correct option.

In each turn of dialogue, you must first determine: Based on the question description, the option descriptions, the currently available patient information, and your own knowledge, do you have enough confidence to select the correct option?

1. If you are not confident enough, output a specific question in the following format:
   Question: [The specific question you want to ask]

2. If you are confident enough, output your selection in the following format: Final Answer: [Your chosen option]

Important Notes:

1. In each turn of conversation, you must make a clear decision — either choose an option or ask a question. Do not be vague. When responding or asking, you must strictly follow the corresponding format.

2. When choosing an option, you can only choose one from the provided options (e.g., A, B, C, etc.), and cannot choose multiple or include any other content.

3. When asking a question, you can only ask one specific question at a time, cannot repeat questions that have already been asked, and cannot include any other content.

4. Interaction Limit: You have a maximum of 8 turns. This means you can ask at most 7 questions and must provide your Final Answer by the 8th turn at the latest.

Initial information: [initial patient information]

Question: [question]

Options: [options]

Your response:

---

**Atomic Facts Extraction Prompt**

You are an honest and knowledgeable medical assistant. I will provide a medical diagnosis case in JSON format, including relevant patient information and the medical problem that needs diagnosis. Your task is to analyze the patient information and extract relevant details. Your final output must be a single, perfectly formatted JSON object.

Extraction Steps

1. Identify the main issue or medical problem described in the information that requires diagnosis, and record it as the "question".

2. Excluding the "question" information, extract a series of key facts from the remaining information. Each fact should contain only one relevant information point and must be self-contained.

3. From the extracted key facts, write a short introductory sentence summarizing the background of the case as "context". Be careful — the context should include as few atomic_facts as possible, no more than two.

4. The remaining key facts should be documented as "atomic_facts".

Output Format Requirements

1. Your only output must be a complete JSON object written in English.

2. The extracted "question" information must be placed in the "question" field, the extracted "context" information must be placed in the "context" field, and the extracted "atomic_facts" information must be placed in the "atomic_facts" field.

3. Output structure example:

```
{
    "question": extracted question,
    "context": [extracted context],
    "atomic_facts": [atomic_fact1, atomic_fact2, atomic_fact3, ...]
}
```

Please start extracting information based on the following input: [input]

---

**User Simulator Quality Monitoring Prompt**

You are given:

1. A list of atomic facts (ground truth facts).
2. A multi-turn interaction between a user and an assistant (called "output" below).

Evaluation criteria:

1. user_answered_related_q: "yes" if the assistant asked a question that is clearly related to any fact in the atomic facts list and the user provided relevant information in their answer. If the assistant's question is unrelated to atomic facts, and the user responds with something like "The patient cannot answer this question.", this is acceptable and should still be marked "yes". Only mark "no" if the assistant's question is strongly related to atomic facts, the user could have answered based on them, but did not provide such an answer.

2. user_hallucinated: "yes" if the user's answer includes fabricated content not appearing in the atomic facts list; "no" otherwise.

Only output a JSON object with exactly these two keys:
```
{
    "user_answered_related_q": "yes" or "no",
    "user_hallucinated": "yes" or "no"
}
```

Do not output anything else.

Atomic facts: [atomic_facts]

Conversation output: [output_text]

---

## A.9 ATOMIC FACTS DATA EXAMPLE

**Atomic Facts Data Example**

1. The symptom is precipitated in the morning.
2. The symptom is precipitated during exams.
3. There is no history of loss of consciousness.
4. Her cousin sister has been diagnosed with epilepsy.
5. An EEG was performed and was suggestive of epileptic spikes.

