# OpenReview forum: "ATPO: ADAPTIVE TREE POLICY OPTIMIZATION FOR MULTI-TURN MEDICAL DIALOGUE"
_ICLR.cc/2026/Conference — ICLR 2026 Poster_

### Official Review · Reviewer_PxMc · 2025-10-25

**Soundness:** 3
**Presentation:** 3
**Contribution:** 3
**Rating:** 8
**Confidence:** 3

**Summary:**

This paper proposed an adaptive tree policy learning method for multi-turn dialogue systems. It builds a conversation tree to calculate the value of nodes and uses reinforcement learning algorithms to improve the system with feedbacks. An uncertainty-aware tree expansion component is also introduced to expand the conversation tree. Experiments show performance improve on different datasets and different large language models.

**Strengths:**

1. The designs of the framework seem reasonable, because conversation has different flows and can be modeled as a tree.\
2. The presentation of the paper is clear.
3. Experiments demonstrate performance improvement on different datasets and different models.

**Weaknesses:**

1. It will be better if other aspects of conversation quality, such as informativeness and helpfulness can be evaluated.
2. The paper claims an improvement of computation cost with KV cache techniques, so a detailed analysis on efficiency will be better.

**Questions:**

This methods seem more general. Can it be applied to general conversation settings other than the medical domain?

---

> ### Author Response · Authors · 2025-11-18
>
> We sincerely thank you for your insightful and valuable feedback on our work. We have carefully studied your comments and hope that the following responses can address your concerns.
>
> * **Regarding Weakness (1):** The experiment in our paper is a task-oriented multi-turn dialogue scenario, where the goal is to train a model to effectively collect key information through multi-turn dialogue to complete a specific downstream task (medical QA in this paper), given incomplete initial information. In this scenario, the model can only achieve high accuracy by collecting key information through high-quality dialogue. Therefore, **we used QA accuracy as an objective metric to measure dialogue quality**. Your comment made us realize that supplementing with other evaluation metrics for dialogue quality would make our paper more complete. In this context, a crucial metric for evaluating dialogue quality is informativeness, i.e., whether the model can ask effective questions to gather sufficient information. Therefore, **in the Appendix A.4**, we have added the change curve of the **proportion of effective questions proposed by the assistant** in multi-round dialogues as the ATPO training progresses,
> along with a concrete case study. It can be seen that as training progresses, the proportion of effective questions asked by the assistant continuously increases, indicating that the assistant's questions are more often answerable by the user. This enables the assistant to collect more useful information and complete tasks with fewer dialogue turns, thereby improving the overall user experience. Therefore, this supplement can **demonstrate the improvement of ATPO on dialogue quality from another perspective**, making our evaluation more comprehensive.

---

> > ### Author Response · Authors · 2025-11-18
> >
> > * **Regarding Weakness (2):** Thank you for your insightful suggestion. To provide a detailed analysis of the computational efficiency improvements brought by our tree-based method, we offer the following theoretical derivation (the derivation process references the implementation of computation statistics in VeRL [1]) and experimental evidence: For an LLM, its internal computational cost can be mainly divided into two parts: the computation of linear projections and the computation of attention mechanism. For an LLM of a specific size, let's assume the number of parameters for linear projections is ${\phi}$ and the number of parameters for attention is ${\theta}$. For a specific turn in the multi-turn dialogue, we denote the input length of the current turn as ${x}$ and the output length as ${y}$. If we sample one data sample at a time, by utilizing the KV cache, we can perform a prefill on the input, with a computational cost of $2{\phi}{x} + 4{\theta}{x}^2$. During decoding, we can reduce computation by using the cached $K$ and $V$, with a computational cost of $2{\phi}{y} + 2{\theta}({2{x}+{y}+1})$. If we sample $N$ data samples of the same length, the total computational cost is $N*(2{\phi}{x} + 4{\theta}{x}^2) + N*(2{\phi}{y} + 2{\theta}({2{x}+{y}+1}))$. However, if we sample $N$ data samples of the same length in a given turn via tree expansion, we can perform the prefill step just once and then decode the $N$ samples. The total computational cost in this case is $1*(2{\phi}{x} + 4{\theta}{x}^2) + N*(2{\phi}{y} + 2{\theta}({2{x}+{y}+1}))$. **Compared to independently sampling $N$ data samples, it can save approximately $(N-1)*(2{\phi}{x} + 4{\theta}{x}^2)$ in computational cost.** Since our proposed ATPO also adopts a tree-structured sampling strategy, it accordingly leads to computational savings. TreePO [2] has already provided rigorous experimental evidence (Section 4.1, Sampling Efficiency Analysis) demonstrating that tree-structured sampling yields higher throughput. Therefore, we do not repeat these experiments and instead present a theoretical analysis of the computational savings in our multi-turn scenario. In contrast to TreePO, where inference chains are segmented for tree-based sampling, our scenario naturally forms semantically clear tree nodes, making it inherently well suited for efficient tree search. We will include this theoretical analysis in the **Appendix A.7** of the revised paper for completeness.
> >
> >
> > * **Regarding Questions:** Certainly! Although the scenario in our paper is multi-turn medical dialogue, in practice, our proposed **ATPO can be extended to various multi-turn dialogue tasks and even multi-turn non-dialogue tasks (e.g., multi-turn tool calling)**. In these scenarios, ATPO can sample high-quality data through its adaptive tree search mechanism, enabling the model to converge faster and more stably. Furthermore, although the final task in our paper is to complete medical QA (where a reward can be quickly given based on the correctness of the answer), it is also applicable to open-ended scenarios. For example, by using the "LLM-as-a-Judge" [3] approach to have another, more capable model judge the model's output and thus provide an appropriate reward.
> >
> > We hope our responses above have addressed your concerns. We recognize the shortcomings in our manuscript and have made revisions, adding the aforementioned experiments and analyses to make our paper more complete.
> >
> > [1] Sheng, Guangming, et al. "Hybridflow: A flexible and efficient rlhf framework." Proceedings of the Twentieth European Conference on Computer Systems. 2025.
> >
> > [2] Li, Yizhi, et al. "Treepo: Bridging the gap of policy optimization and efficacy and inference efficiency with heuristic tree-based modeling." arXiv preprint arXiv:2508.17445 (2025).
> >
> > [3] Li, Dawei, et al. "From generation to judgment: Opportunities and challenges of llm-as-a-judge." Proceedings of the 2025 Conference on Empirical Methods in Natural Language Processing. 2025.

---

> > > ### Comment · Reviewer_PxMc · 2025-11-19
> > >
> > > Thank you for your response, which has addressed my concerns. I will maintain my positive rating.

---

> > > > ### Author Response · Authors · 2025-11-21
> > > >
> > > > We sincerely thank you for taking the time to review our paper and for your positive feedback on our latest responses. We are glad that our additional experiments and theoretical analyses have addressed your concerns. We appreciate your constructive suggestions, which have helped us improve the completeness and clarity of our work.

---

### Official Review · Reviewer_Pcyh · 2025-10-29

**Soundness:** 2
**Presentation:** 3
**Contribution:** 3
**Rating:** 4
**Confidence:** 4

**Summary:**

This paper proposes a new RL training method to improve LLMs ability at solving medical multi-turn dialogues, which involves asking many (clarification/diagnosis) questions in multiple turns before returning the final answer. The authors identify that a key challenge in optimizing multi-turn dialogue is the high uncertainty in user's responses, resulting in often insufficient exploration/data coverage for training a robust enough medical agent. The proposed method (ATPO) aims to solve this by utilizing tree search algorithms, so that during RL rollout ATPO prioritizes on expanding nodes with high uncertainty. During optimization, ATPO then treats each branch of the search tree as an independent trajectory to perform loss computation/backpropagation. The authors experimented on three medical benchmarks and showed improved performance of ATPO compared to baselines such as GRPO and TreePO.

**Strengths:**

- The notion to expand nodes and allocate more rollout budget to states with high uncertainty is sound. I believe the overall design (with some abstractions) could be applicable to other dialogue tasks beyond medical benchmarks.

- The authors provided evaluation on three different medical benchmarks, showing improvement of ATPO compared to other training methods such as GRPO and TreePO.

- The authors also provided interesting analysis of performing tree-based training during RL. Specifically, the authors experimented with using different uncertainty metrics during RL/the importance of down-weighing policy updates by node visit counts for stability.

**Weaknesses:**

Despite the overall positive experimental results, I believe there are some uncertainty/flaw in the experimental setup that could substantially undermine the results and comparisons made. If these are addressed I am willing to increase my soundness and overall score. I detail them below.


1. Tree based methods generally requires much more compute compared to methods such as PPO/GRPO, as they need to perform multiple policy and value inference per state. However, neither Table 1 nor Figure 2 reports FLOPs statistics or # of LLM calls during rollout, raising concerns that the observed gains may stem from simply using more compute rather than true algorithmic improvements. I suggest the author to (1) report FLOPs used during rollout stage across different methods (especially comparing search based method such as ATPO and non-search based method such as GRPO and PPO), and (2) provide compute-equal comparisons between ATPO and methods such as GRPO and PPO (e.g., if ATPO uses N flops during the entire rollout stage, adjust the batch size and group size of GRPO so that it also uses N flops during rollout).


2. Many results in Table 1 are actually improvements *within one standard deviation*. For example, the MedQA and MedMCQA result on Qwen3-1.7B (TreePO vs ATPO), MedMCQA result on Qwen3-4B, and MedicalExam and MedMCQA result on Qwen3-8B. Additionally, simply prompting GPT-4o already achieves performance within one standard deviation of the best results across all benchmarks. Could the authors provide a statistical significance analysis to confirm whether these improvements are indeed meaningful?


3. Since simply prompting GPT-4o achieves near best result (within one standard deviation) in Table 1 across all benchmarks, I wonder why did the author not compare against a simple baseline of directly distilling (correct) dialogue outputs obtained by prompting GPT-4o? Instead, this paper uses self-play data obtained from Gemini-2.5-Pro (L321-323), whose performance is also not reported in Table 1. This is because (1) improvements from RL tends to heavily depends on the quality of the initial SFT checkpoint, which may be undertrained in this work; (2) distilling outputs from GPT-4o is far more computationally efficient than SFT + RL, making it a more practical choice in the context of this paper if prompting GPT-4o already achieves near-best performance.

**Questions:**

- Have the authors measured test performance when a user simulator *different from training* is used (e.g., LLaMA-3.3 or LLaMA-4 models). It is likely that RL training with a fixed user simulator may overfit to specific tone/response formats of the user simulator used during training, and may not generalize to other unseen users/user simulators at test time.

---

> ### Author Response · Authors · 2025-11-18
>
> We sincerely thank you for your insightful and valuable feedback on our work. We have carefully studied your comments and hope that the following responses can address your concerns.
>
> * **Regarding Weakness (1):** We fully understand your concern that the lack of detailed analysis on computational resources in our manuscript might lead readers to misinterpret ATPO's improvements as merely a result of using more compute. Therefore, we have added a detailed analysis of computational resources in **Appendix A.5**. Specifically, we measured the time each algorithm needed to reach the same test-set accuracy (in this case, the best performance achieved by PPO). The results are summarized in the table below. As shown, ATPO required the least time to achieve the target performance, while GRPO took the longest. Given that all algorithms were run on identical computing hardware, a shorter time means lower consumption of computing resources. Thus, **ATPO's optimal results are not achieved by using more computational resources but are attributed to its effectiveness**. Furthermore, in our experiments, we observed that with continued training, PPO's performance curve had already converged with no further improvement, whereas GRPO and TreePO even experienced entropy explosion and ultimately crashed. We also measured the proportion of total runtime spent in the rollout stage for each algorithm, along with the distribution of computation within this stage, as shown in the table below. ATPO exhibited the highest rollout time proportion, accounting for approximately 45% of the total runtime, while the other methods had nearly identical proportions of around 25%. Notably, more than half of ATPO's rollout-stage computation is devoted to value estimation, which is reasonable given that ATPO frequently evaluates node values during rollout to dynamically allocate its rollout budget. However, **these costs are meaningful**, as it produces high-quality sampling data, thereby accelerating model convergence. This is corroborated by Figure 2(a) of the manuscript, where ATPO's convergence curve is steeper than those of other algorithms. In contrast, for other methods, rollout-stage computation is dominated by sequence generation, producing lower-quality sampled data that slows convergence. In summary, while ATPO allocates more time to the rollout stage, its emphasis on value estimation produces higher-quality data, resulting in faster convergence, reduced total training time, and ultimately lower overall computational resource consumption compared to competing algorithms.
>
>     | Method | Total Runtime (hour) | Proportion of Rollout Time | Rollout TFLOPs | Proportion of Genration TFLOPs | Proportion of Value Estimation TFLOPs |
>     | :--- | :--- | :--- | :--- | :--- | :--- |
>     | PPO | 3.02 | 26.16% | 444,915.23 | 100% | 0% |
>     | ATPO ($U_1$+$U_2$) | 2.22 | 45.05% | 639,410.69 | 37.99% | 62.01% |
>     | ATPO ($U_1$) | 2.98 | 41.95% | 817,173.44 | 42.38% | 57.62% |
>     | TreePO | 3.15 | 24.13% | 916,408.62 | 100% | 0% |
>     | GRPO | 4.86 | 25.72% | 4,708,526.96 | 100% | 0% |
>
>
> *   **Regarding Weakness (2):** Thank you for your valuable comment. To assess the significance of the improvements, we conducted a two-tailed independent samples t-test comparing ATPO ($U_1$+$U_2$) against TreePO across different model sizes and test sets, as well as Qwen3-8B trained with ATPO versus GPT-4o with prompting. The results are presented in the table below, where bolded p-values (<0.05) indicate significant differences. From the table, we observe that ATPO does not significantly outperform TreePO or GPT-4o in most cases, but shows clear gains in three instances: Qwen3-4B on MedicalExam, Qwen3-4B on MedQA, and Qwen3-8B on MedQA. This aligns with expectations—**ATPO's main advantage lies in efficiency**. As noted in the response to Weakness (1), **ATPO achieves comparable performance to TreePO with fewer training steps and lower computational cost**. Furthermore, being able to train an 8B-parameter model (Qwen3-8B) to match the performance of GPT-4o demonstrates ATPO's effectiveness. While the gains are often not statistically significant, **ATPO offers notable efficiency benefits and competitive performance given the model scale and resources used**.
>
>
>     | Model | Test dataset | vs. TreePO | vs. GPT-4o |
>     |---|---|---|---|
>     | Qwen3-1.7B | MedicalExam | 0.9165 | / |
>     | Qwen3-1.7B | MedQA | 0.1948 | / |
>     | Qwen3-1.7B | MedMCQA | 0.2004 | / |
>     | Qwen3-4B | MedicalExam | **0.0333** | / |
>     | Qwen3-4B | MedQA | **0.0131** | / |
>     | Qwen3-4B | MedMCQA | 0.2676 | / |
>     | Qwen3-8B | MedicalExam | 0.8118 | 0.3932 |
>     | Qwen3-8B | MedQA | **0.0026** | **0.0421** |
>     | Qwen3-8B | MedMCQA | 0.3641 | 0.4340 |

---

> > ### Author Response · Authors · 2025-11-18
> >
> > * **Regarding Weakness (3):** Thank you for your valuable feedback. A good initial SFT checkpoint is indeed important for the subsequent RL. However, the importance of SFT lies in teaching the model to ask questions and provide answers in a specified format. For example, it teaches the model to place the thought process within `<think></think>`, the question after `Question:`, and the final answer after `Final Answer:`. This facilitates subsequent rewarding or penalizing of the model's behavior. Therefore, this stage only requires a model that can effectively simulate multi-turn dialogue data that meets the format requirements; the specific model used does not have a significant impact. In our paper, we used multi-turn dialogue data simulated by Gemini-2.5-Pro for training, but due to an oversight, we did not report the performance of Gemini-2.5-Pro with direct prompting on the test sets. We have therefore added the corresponding results, as shown below.
> >     | Model | Method | Method Name | MedicalExam | MedQA | MedMCQA |
> >     |---|---|---|---|---|---|
> >     | GPT-4o | Prompt | MEDIQ | $64.00\pm3.53$ | $63.15\pm0.82$ | $53.03\pm0.89$ |
> >     | Gemini-2.5-Pro | Prompt | MEDIQ | $74.33\pm2.53$ | $68.69\pm0.61$ | $63.31\pm1.37$ |
> >
> >     It can be observed that Gemini-2.5-Pro, being a more powerful model, indeed achieves better performance than GPT-4o (although the ATPO-trained Qwen3-8B does not surpass Gemini-2.5-Pro's performance, this is reasonable given its parameter size of only 8B). However, to demonstrate that the choice between these two models does not significantly affect the SFT training outcome and to show that the SFT stage is sufficiently trained, we used the same prompts to have both GPT-4o and Gemini-2.5-Pro strictly simulate dialogues based on the data from the RL training dataset (14,256 samples). Then, we perform SFT on the Qwen3-8B model separately using the dialogue data generated by the two models (i.e., distilling the correct dialogue outputs from GPT-4o and Gemini-2.5-Pro separately), and test the performance. The results are shown in the table below. As can be seen, even with more data and longer training, **the model's performance shows almost no improvement.** This further validates that the importance of the SFT stage is to teach the model the format for asking and answering questions, that it does not require a particularly large amount of data to be sufficiently trained, and that the choice of model for dialogue simulation is not critical as long as the simulated data meets the format requirements. Moreover, **through SFT training alone, even with a large amount of data, the model still lacks generalization ability, making RL training still necessary for improving its generalization.**
> >
> >     | Model | Dataset | Dialogue Simulator | MedicalExam | MedQA | MedMCQA |
> >     | :--- | :--- | :--- | :--- | :--- | :--- |
> >     | Qwen3-8B | MEDIQ validation dataset | Gemini-2.5-Pro | $55.87\pm0.30$ | $53.75\pm1.18$ | $46.87\pm1.74$ |
> >     | Qwen3-8B | RL training dataset | GPT-4o | $57.33\pm2.26$ | $53.25\pm0.72$ | $46.04\pm1.31$ |
> >     | Qwen3-8B | RL training dataset | Gemini-2.5-Pro | $58.67\pm0.08$ | $51.51\pm1.33$ | $48.06\pm0.94$ |
> >
> > * **Regarding Questions:** Thank you for your very insightful comment. Since we used Qwen3-8B to simulate the user during RL training, it is indeed possible that the trained model might overfit to the specific tone/response format of the user simulator used during training. To test this, as you suggested, we replaced the user simulator at test time with the Llama-3.3-70B-Instruct model and re-evaluated the performance of Qwen3-1.7B, Qwen3-4B, and Qwen3-8B trained with ATPO ($U_1$+$U_2$). The results are shown below. As can be seen, **after changing the user simulator, assistant's performance is almost identical to before, with no significant difference**. This indicates that assistant model has not overfit to the specific tone/response format of the user simulator used during training and performs well on an unseen user simulator, which fully demonstrates its good generalization capability.
> >     | Model | User Simulator | MedicalExam | MedQA | MedMCQA |
> >     | :--- | :--- | :--- | :--- | :--- |
> >     | Qwen3-1.7B | Qwen3-8B | $43.20\pm1.85$ | $42.87\pm0.77$ | $39.93\pm1.05$ |
> >     | Qwen3-1.7B | Llama-3.3-70B-Instruct | $43.07\pm1.53$ | $43.15\pm0.60$ | $40.07\pm1.39$ |
> >     | Qwen3-4B | Qwen3-8B | $59.73\pm2.61$ | $55.47\pm0.99$ | $45.93\pm1.13$ |
> >     | Qwen3-4B | Llama-3.3-70B-Instruct | $61.20\pm2.76$ |$ 56.47\pm0.94$ | $47.76\pm1.06$ |
> >     | Qwen3-8B | Qwen3-8B | $65.87\pm3.72$ | $64.07\pm0.43$ | $53.66\pm1.52$ |
> >     | Qwen3-8B | Llama-3.3-70B-Instruct | $67.07\pm3.35$ | $63.93\pm1.01$ | $55.60\pm1.21$ |
> >
> > We hope our responses above have addressed your concerns. We recognize that some key analyses were missing from our manuscript, and we have since made revisions, adding the key supplementary experiments mentioned above to make our paper more complete.

---

> ### Comment · Reviewer_Pcyh · 2025-11-20
>
> Thanks for the detailed response.
>
> Regarding Weakness 1, I believe the authors are arguing although ATPO requires more compute per training step (e.g., more rollouts), it converges faster compared to other algorithms such as GRPO/PPO. The additional table provided by the authors demonstrate this by presenting the time needed to reach *PPO's best performance*, but its unclear why PPO is chosen in this case, as it is amongst the weakest according to your Table 1? *Could you please provide comparison against GRPO (the strongest non-tree-search method in Table 1) instead of PPO, which I think would make your results more convincing.*
>
> Regarding Weakness 2, I believe this is mostly resolved.
>
> Regarding Weakness 3, the authors show that even though (e.g.,) Gemini-2.5-Pro achieves 74.33, 68.69, 63.31 on MedicalExam, MedQA, MedMCQA respectively, distilling outputs from Gemini-2.5-Pro to Qwen3-8B still significantly underperforms and can only achieve 58.67, 51.51, 48.06. I believe this is possible, as I have seen many other work showing that distillation often underperforms due to model capability/task is too challenging. I thus believe my concern is mostly resolved, though I strongly encourage the authors to add this comparison to Table 1, because distillation from strong LLMs should always be an easy yet strong baseline.
>
>
> ---
>
> Overall, I think weakness 1 is still unresolved. However, I have increased the soundness and overall score to 6, assuming result on comparing against GRPO would also hold, given your claim that ATPO generally converges/learns faster.

---

> > ### Author Response · Authors · 2025-11-21
> >
> > Thank you very much for your positive feedback on our previous response.
> > * **Regarding Weakness (1)**: In the supplementary experiments, we chose the best performance of PPO as a unified target. The reason for this choice is that we intended to compare all algorithms under the same target. If the target were set too high, PPO—whose training curve had already converged—would be unable to reach it even with extended training, making it impossible to accurately measure the computational resources required for PPO to achieve that accuracy.
> >
> >     To provide further clarification, we further computed, for all algorithms (excluding PPO for the reason above), the time required to reach the same accuracy on the test set (set here to the best performance of GRPO), the proportion of that time spent in the rollout stage, as well as the computation and its detailed distribution in the rollout stage. The results are shown in the table below.
> >
> >     It can be observed that ATPO still requires the least amount of time, which indicates that it also demands the least computational resources. Furthermore, from the proportion of time and the computation distribution in the rollout stage, we reach the same conclusion as before: although ATPO spends more time in the rollout stage, the higher quality of the sampled data enables faster model convergence, resulting in a reduced total time and, consequently, fewer resources needed overall.
> >
> >     We hope that these newly added results address your remaining concerns, and we sincerely appreciate your constructive feedback once again.
> >
> >
> >     | Method | Total Runtime (hour) | Proportion of Rollout Time | Rollout TFLOPs | Proportion of Genration TFLOPs | Proportion of Value Estimation TFLOPs |
> >     | :--- | :--- | :--- | :--- | :--- | :--- |
> >     | ATPO ($U_1$+$U_2$) | 2.35 | 47.73% | 668,801.98 | 38.02% | 61.98% |
> >     | ATPO ($U_1$) | 3.17 | 41.49% | 848,271.74 | 42.58% | 57.42% |
> >     | TreePO | 3.68 | 27.52% | 1,016,754.67 | 100% | 0% |
> >     | GRPO | 5.41 | 25.19% | 5,174,142.08 | 100% | 0% |
> >
> >
> >
> > * **Regarding the Distillation Experiment**: We sincerely appreciate your valuable suggestion regarding the distillation experiment. Thanks to your feedback, we have conducted and included this important baseline experiment in our revised manuscript. In addition, we have expanded our discussion to cover the role of the SFT stage, the amount of data required during this stage, and the choice of the model in constructing SFT data, which we believe offers further insights for readers. While we initially intended to present the distillation results in Table 1, due to page limitations we have placed them in Appendix A.3. We have included a clear link in the main text so that readers can easily access the results and our analysis. We hope you understand this arrangement, and we are truly grateful for your thorough review and constructive feedback.

---

> > > ### Comment · Reviewer_Pcyh · 2025-11-24
> > >
> > > Thanks for the additional response and results. I believe my concerns are mostly resolved. I have already updated my scores and will keep the overall positive rating.

---

> > > > ### Author Response · Authors · 2025-11-25
> > > >
> > > > We sincerely thank you for taking the time to review our paper and for your positive feedback on our latest responses. We are glad that our additional experiments have addressed your concerns. We appreciate your constructive suggestions, which have helped us improve the completeness and clarity of our work.

---

### Official Review · Reviewer_8sJu · 2025-10-31

**Soundness:** 1
**Presentation:** 2
**Contribution:** 3
**Rating:** 2
**Confidence:** 4

**Summary:**

This paper proposes a framework, namely ATPO, for generating multi-turn medical dialogues, which enhances the performance of LLMs in medical QA datasets.

**Strengths:**

(1) This paper has some theoretical derivation for the framework.

**Weaknesses:**

1) However, I find some parts of the theoretical derivation questionable. 1） When defining the Bellman error in equations (1) and (2), the authors use a one-step lookahead. This is questionable since with one step, there is no long-term reward (i.e., long-term exploration). In this case, equations (1) and (2) would collapse to the average reward of all the states. If the authors finally generate an answer based on the whole dialogue, it makes more sense to enable the Bellman error to look multiple steps. 2) When writing equations (1) and (2), the authors need to freeze one of the critics to calculate the Bellman error. I didn't see any mention of this.
(2) This is a fundamental question. The authors are using LLM-generated data to train LLM. This can be problematic. For example, the authors used Gemini-2.5-Pro to simulate the conversation data. There will be data leakage if the Gemini-2.5-Pro also uses the same dataset for its pretraining or fine-tuning. Thus, the improvements in Table 1 can be attributed to the data leak.
(3) The title of this paper talks about the optimization for multi-turn medical dialogue. But there is no evaluation of the quality of the generated dialogue! Instead, the authors use medical QA performance to justify the usefulness of the multi-turn dialogue. The authors need to think clearly about this. If your purpose is to have a better performance on medical QA, then you need to compare with the SOTA methods. If the purpose is to generate better multi-turn medical dialogue, you need to improve Cons (1) and (2).

**Questions:**

I encourage the authors to keep polishing this work. If the authors can improve or convince others about the Cons (1) and (2), and reorganize the paper, I believe this can be a good work.

---

> ### Author Response · Authors · 2025-11-18
>
> We sincerely thank you for your insightful and valuable comments on our work. We have carefully studied your comments and hope the following reply can address your concerns.
>
> *   **Regarding Weakness (1) [Theoretical Derivation]:**
>     *   **1) On the "short-sightedness" of one-step lookahead:** We fully understand your concern. Our task involves multi-turn dialogue, which is a long-horizon task, yet we only use a one-step lookahead in equation (1). You are concerned that this might cause our algorithm to lack long-term planning, leading to "short-sightedness" (i.e., being unable to optimize long-term dialogue quality). Here, we would like to clarify that the reason we use a one-step lookahead is that we want to quickly decide which node with the highest uncertainty to prioritize for exploration during tree expansion process, thereby optimizing the allocation of the rollout budget. Using a multi-step lookahead would cause a significant increase in time consumption, so we chose not to use multi-step lookahead. Therefore, we proposed the uncertainty metric $U$, which combines $U_1$ and $U_2$, to guide the tree expansion process. It is important to note that there is no need to worry about this causing the model to be "short-sighted" because, as described in Section 3.2, each node $x_k$ represents the dialogue history up to the current turn. $V(x_k)$ evaluates the value of the information gathered in these dialogues for completing the final task. **A higher value indicates a higher value of the dialogues, which is more conducive to ultimately completing the task.** Thus, it does not lead to a "short-sightedness" problem. Furthermore, as shown in equation (5), the update target for our $V(x_k)$ is the **true cumulative reward** obtained during the tree expansion sampling process. Updating the critic model towards this target also allows the critic model to more accurately estimate the true value, which in turn better guides the tree expansion process.
>     *   **2) On freezing the critic network:** We are sorry for the confusion caused by our unclear description. Let's clarify it. Unlike reinforcement learning algorithms that involve bootstrapping, such as DQN [1] and SAC [2], ATPO is an on-policy actor-critic reinforcement learning algorithm, and its critic model training **does not involve bootstrapping**. In bootstrapping updates, the update target itself contains the output of the neural network. Consequently, the target changes continuously as the network parameters are updated, which may cause instability in the neural network's training. To solve this problem, the concept of a target network is often used: the network within the target is temporarily frozen. However, Equations (1) and (2) in our paper correspond to the tree expansion process. This process **only calls the critic model to estimate the value of the nodes and does not involve training the critic model.** In other words, **the Bellman error here is not used as the loss function for training the critic model**. The actual update target for the critic is the real cumulative reward obtained during the sampling process, as shown in Equation (5) in our paper. **The actual loss function for training the critic model is as shown in Equation (8)** (note that $\hat{V}$ represents the value estimated from the real reward, while $V$ represents the value estimated by the critic model). This means we are not updating the critic model in a bootstrapping manner, so there is no issue of training instability caused by the update target being the output of the neural network itself. Since Equations (1) and (2) do not involve critic model training at all, so it is not necessary to specify which critic model is frozen.
>
> [1] Mnih, Volodymyr, et al. "Human-level control through deep reinforcement learning." nature 518.7540 (2015): 529-533.
>
> [2] Haarnoja, Tuomas, et al. "Soft actor-critic: Off-policy maximum entropy deep reinforcement learning with a stochastic actor." International conference on machine learning. Pmlr, 2018.

---

> > ### Author Response · Authors · 2025-11-18
> >
> > *   **Regarding Weakness (2) [Data Leakage]:** We understand your concern that using multi-turn conversations between a user and an assistant, simulated by Gemini-2.5-Pro, for SFT training could lead to data leakage—because Gemini-2.5-Pro might have been trained on our test sets—which would in turn cause an fake improvement of the results in our Table 1. However, we would like to clarify that the purpose of our SFT training here is not to leak knowledge to the model, but rather to teach the model to follow a specific format for asking and answering questions. For example, teach the model to place the thought process within `<think></think>`, the question after `Question:`, and the final answer after `Final Answer:`. In this process, we prompt Gemini-2.5-Pro to strictly simulate conversations based on the data from the MEDIQ validation dataset (1,272 entries). The prompt is available in the code repository because it is too long to display in the paper, and we have conducted thorough checks to ensure the generated dialogues strictly follow the provided data. Our testing, on the other hand, was conducted on the MedQA test dataset, the MedMCQA test dataset, and the MedicalExam test dataset. **These three test datasets have no overlap with MEDIQ validation dataset.** Furthermore, as can be seen from the results in Table 1 of the paper, the performance of the model trained with SFT (Qwen3-1.7B on MedicalExam, MedQA, MedMCQA, and Qwen3-4B on MedMCQA) on the test sets even showed a slight decrease compared to direct answering. **If data leakage had truly occurred, it would be reasonable to expect an increase in these results.** To provide further evidence, we used the same prompts to have both GPT-4o and Gemini-2.5-Pro strictly simulate conversations based on the data from the RL training dataset (14,256 entries). Then, we perform SFT on the Qwen3-8B model separately using the dialogue data generated by the two models, and test the performance. The results are shown in the table below, with the first row displaying previous results for easy comparison. As can be seen, even with additional data and extended training time, the model's performance only shows a marginal improvement or even a slight decrease compared to previous results (on MedQA). If data leakage were an issue, training with more data should logically lead to better performance, but this is not the case. **This further demonstrates that the performance improvement of the models shown in Table 1 does not stem from data leakage.**
> >
> >     | model | Dataset | Simulator | MedicalExam | MedQA | MedMCQA |
> >     |---|---|---|---|---|---|
> >     | Qwen3-8B | MEDIQ validation dataset | Gemini-2.5-Pro | $55.87\pm0.30$ | $53.75\pm1.18$ | $46.87\pm1.74$ |
> >     | Qwen3-8B | RL training dataset | GPT-4o | $57.33\pm2.26$ | $53.25\pm0.72$ | $46.04\pm1.31$ |
> >     | Qwen3-8B | RL training dataset | Gemini-2.5-Pro | $58.67\pm0.08$ | $51.51\pm1.33$ | $48.06\pm0.94$ |

---

> > > ### Author Response · Authors · 2025-11-18
> > >
> > > *   **Regarding Weakness (3) [Mismatched Evaluation and Motivation]:** Thank you very much for your insightful thoughts on our paper's positioning. We acknowledge that the explanation of the motivation and evaluation logic in our manuscript may not have been clear enough, leading to your confusion. It needs to be clarified that the ATPO we proposed is to solve a specific and practical problem: in the case of incomplete initial information, how to train a model to effectively collect key information through multi-turn dialogue to complete a well-defined downstream task (medical QA in this paper). In this task-oriented setting, the quality of the dialogue directly affects whether the key information needed to solve the problem can be successfully obtained. That is, the reason we use medical QA performance to justify the usefulness of the multi-turn dialogue is that, in this experimental scenario, if the model cannot effectively gather useful information from previous dialogue turns with the user, it will ultimately be unable to correctly select the right option (for example, the performance of the model using the Direct method in Table 1 is poor). A specific example is shown in Figure 3 of the paper, where the model can only answer correctly by gathering user background information and eliminating wrong options through high-quality dialogue. Therefore, **QA accuracy is the a objective metric for measuring the quality of information-seeking dialogue in our specific scenario.** To provide a more complete evaluation of our algorithm for dialogue quality improvement, we have added to the **Appendix A.4** a curve showing the change in **the proportion of effective questions** (questions that elicit valid user responses) asked by the assistant in a multi-turn dialogue as ATPO training progresses, as well as a specific case study. It can be seen that as training proceeds, the proportion of effective questions posed by the assistant continuously increases, indicating that the assistant learns to gather effective information through high-quality questioning, thereby better completing the task. We hope these supplements will resolve your confusion and make our manuscript more complete.
> > >
> > > We realize there are some unclear descriptions in our manuscript. We will optimize these parts in the revised version to make the logic clearer and easier for readers to understand, and we will also add the supplementary experiments mentioned above. Once again, we sincerely thank you for your valuable and constructive comments!

---

> > ### Comment · Reviewer_8sJu · 2025-11-20
> >
> > Thanks for the response from the authors. Due to the potential data leakage issue as discussed in weakness (2), I have to keep my rating.

---

> > > ### Author Response · Authors · 2025-11-24
> > >
> > > Thank you very much for your feedback and suggestions on our work. In response to your concerns regarding potential data leakage, we provided relevant additional experiments in our previous reply; however, unfortunately, they did not completely eliminate your concerns. Therefore, to more convincingly verify the reliability of the results in Table 1 and fundamentally rule out the possibility of data leakage, we conducted the following experiments.
> > >
> > > As you pointed out in your previous comments, your main concern arises from our use of Gemini-2.5-Pro in the SFT stage to generate multi-turn conversation data, while Gemini-2.5-Pro may have been exposed to related data during its own training process, which could lead to potential data leakage risks.
> > >
> > > To address this possibility, in the new experiments, **we directly used the model to be trained itself to generate the multi-turn dialogue data required for the SFT stage**. Taking Qwen3-4B as an example, when training this model, we first **used Qwen3-4B itself to simulate the generation of multi-turn dialogue data**, then performed SFT training on this data, and subsequently conducted ATPO training. The experimental results are shown in the table below, and the training curves are provided in the code repository. It can be seen that the model after SFT performs worse compared to directly using the Prompt methods (Direct and MEDIQ), indicating that the model after SFT suffers from weaker generalization and performs worse on the test set. However, **after continuing ATPO training on this SFT model, the performance improves significantly**. This result **fundamentally proves that the performance gains brought by ATPO are indeed due to the effectiveness of the method rather than data leakage**, which also confirms that the improvements in Table 1 are not the result of data leakage.
> > >
> > > It is worth mentioning that when comparing the results of models fientuned with Gemini-2.5-Pro simulated dialogue data followed by ATPO training, our results here are slightly inferior. This is reasonable because the SFT checkpoint can affect the subsequent RL performance, but it should be noted that this is not due to data leakage. Rather, it is because the SFT checkpoint influences the data sampled during the RL stage. For example, Gemini-2.5-Pro simulated dialogues usually have more turns, which causes the SFT model to tend to ask more turns of questions, making it easier to sample high-quality trajectories during RL training, thereby benefiting RL. This has been demonstrated in many related studies [1][2].
> > >
> > > In summary, SFT+RL is an effective training paradigm that has been validated in many previous works [1][2][3], where the SFT checkpoint affects the data sampled in RL and consequently the RL performance. In Table 1, when comparing various RL algorithms, all are based on the same SFT checkpoint, which demonstrates the effectiveness of ATPO compared to other RL algorithms. Moreover, our experiments here show that **ATPO, as an RL algorithm designed to enhance the quality of multi-turn dialogues, can improve model performance on top of different SFT checkpoints**. Therefore, the performance improvement is confirmed to be due to the validity of the ATPO itself, rather than a spurious gain caused by data leakage.
> > >
> > > We hope that these new experiments and results can eliminate your concerns and lead to your recognition of our work. Once again, thank you for your attention to and valuable comments on our research, and we look forward to your reply.
> > >
> > >
> > > | model | Method NAME | MedicalExam | MedQA | MedMCQA |
> > > |---|---|---|---|---|
> > > | Qwen3-4B | Direct | $48.13\pm0.87$ | $44.94\pm0.35$ | $41.53\pm0.39$ |
> > > | Qwen3-4B | MEDIQ  | $45.87\pm1.20$ | $40.11\pm0.60$ | $31.64\pm1.41$ |
> > > | Qwen3-4B | SFT    | $43.60\pm4.15$ | $44.48\pm1.15$ | $38.43\pm1.37$ |
> > > | Qwen3-4B | SFT+ATPO($U_1+U_2$) | $56.40\pm2.14$ | $51.55\pm0.63$ | $45.30\pm0.69$ |
> > >
> > > [1] Guo, Daya, et al. "Deepseek-r1 incentivizes reasoning in llms through reinforcement learning." Nature 645.8081 (2025): 633-638.
> > >
> > > [2] Ding, Hongxin, et al. "Promed: Shapley information gain guided reinforcement learning for proactive medical llms." arXiv preprint arXiv:2508.13514 (2025).
> > >
> > > [3] Feng, Yichun, et al. "Doctoragent-rl: A multi-agent collaborative reinforcement learning system for multi-turn clinical dialogue." arXiv preprint arXiv:2505.19630 (2025).

---

> > > ### Author Response · Authors · 2025-11-27
> > >
> > > Dear Reviewer 8sJu,
> > >
> > > I hope this message finds you well.
> > >
> > > As the discussion period progresses, we would like to kindly ask whether you have had a chance to review our latest supplementary response to your comments. In addressing the main concerns you raised, we have conducted further experiments and clarifications:
> > >
> > > - **Theoretical Derivation Issue**: In our previous response, we have explained in detail the rationale for the one-step lookahead and clarified the question regarding freezing the critic. We have also refined the corresponding descriptions in the manuscript to avoid possible misunderstanding.
> > >
> > > - **Data Leakage Concern**: To completely rule out potential data leakage risks, in the new experiments we directly used the model to be trained itself to generate multi-turn dialogue data, without using Gemini-2.5-Pro at all. The results show that even under these conditions, ATPO still brings significant performance improvements, fundamentally confirming that the gains in Table 1 are not due to data leakage.
> > >
> > > If these additional clarifications and modifications sufficiently address your concerns, we would be very grateful for your recognition. Should you need more technical details, we would be happy to provide them.
> > >
> > > Once again, thank you very much for your valuable attention and constructive feedback on our work.
> > >
> > > Best regards,
> > > The Authors

---

> > > > ### Comment · Reviewer_8sJu · 2025-11-27
> > > >
> > > > The authors have conducted some experiments regarding Cons (1) and (2). I have raised my rating.

---

> > > > > ### Author Response · Authors · 2025-11-27
> > > > >
> > > > > We sincerely thank you for taking the time to review our paper and for your positive feedback on our reply. We are delighted that our additional experiments and analyses have addressed your concerns, and we greatly appreciate your recognition of our work. Your constructive suggestions have been invaluable in improving the completeness and clarity of our manuscript. Once again, thank you for your time, effort, and support.

---

### Official Review · Reviewer_Dv8Q · 2025-11-03

**Soundness:** 3
**Presentation:** 4
**Contribution:** 3
**Rating:** 8
**Confidence:** 4

**Summary:**

This paper addresses sampling for RL models being trained for multi-turn conversations with long-term objectives. Specifically, the background use-case involves answering medical multiple-choice questions in an interactive scenario where the LLMs must ask clarification questions proactively (seek information when missing) before providing the answer.

To this end, the authors extend recent work on Tree Policy Optimization developed for reasoning tasks for information-seeking conversational scenarios. The default TreePO would incur high computational costs to roll-out all possible trajectories from a given “state” for multi-turn conversations while estimating the policy and value functions during RL.

To mitigate these costs, they propose measuring the uncertainty of a state and pruning mechanisms based on this uncertainly thus making the sampling process efficient while ensuring diverse exploration. Their proposed metric combines both the epistemic uncertainty (limitations of the model wrt selection among possible actions from a given state) and aleatoric uncertainty (e.g. induced due to user responses to questions) and  only those branches which meet thresholds on uncertainty are retained resulting in efficient tree expansion during sampling.

Coupled with asynchronous search architectures and KV caches that enable prefix sharing, their experiments show not only improvements in final task (MCQA) accuracy but are also compute efficient. In particular, their performance is on-par with SOTA and slightly out-performs GPT-4o model using a significantly smaller Qwen3-8B model on three different datasets from the medical domain.

**Strengths:**

The paper makes a good extension to recent works on using TreePO for training RL models for reasoning tasks and conversation forests on the task of answering MCQs in medical domain. This is an important task in the important domain of medical diagnosis that may specifically prefer small, in-house models than proprietary ones such as GPT.

The paper is well-written, building up the intuition for the proposed metric as well as the developing equations for actor/policy and critic/value updates.

The main claims are suitably supported via detailed experiments and ablation studies on three separate datasets from the topic.

Overall, a valuable contribution advancing the state-of-the-art in modeling efficiency aspects of sampling in RL for multi-turn conversations.

**Weaknesses:**

Not weaknesses specifically but some areas for improvement where details are lacking or there are clarity issues are listed below--

- Contributions (line 84) --This aspect is not really highlighted much in the paper and as such the setup seems identical to that described in the TreePO paper (Li et al 2025b cited by the authors). If different, please highlight the differences and why this forms a core contribution.

- In general, the performances in Table-1 seem very close to TreePO in many instances—would like to see some explanation on key differences between the two methods (if there is anything else apart from the pruning aspect which save on computation)—the lines 333-338 are not very clear

- Overall discussion on parameters, possible application in practice, and some details on experiments need presentation improvements, some of which are listed in the Questions section.

**Questions:**

- Would like to see some discussion on how this may work in “real world” usage where the MCQ options are not available? How might that look like? How will the process scale when the number of possibilities become 10 instead 4? What are the consequences when the user assistant does not behave “ideally” providing the correct answer in response during inference? May be the above can be discussed using the anecdotal example in Figure 3.

- Some discussion on where the pruning might breakdown, chances of throwing out useful trajectories? Intuition for setting the threshold, how does one do this in practice and why was it set to 0.5/1.5 in your experiments?

- Lines 255-257 seems to be missing some qualification information, as to when the critic value estimates are used -- requires rewriting for clarity.

- For the datasets, where atomic fact extraction was done with Gemini etc (Appendix A.2), include the prompts and other details or provide references, if same as those used for other datasets. What does an atomic fact look like? How do you ensure with GPT-4o that the user assistant is behaving correctly? (305-308)

---

> ### Author Response · Authors · 2025-11-18
>
> We sincerely thank you for your insightful and valuable comments on our work. We have carefully reviewed your comments and hope the following reply can address your concerns.
>
> *   **Regarding Weakness (1) [Emphasis on contributions]**: Thank you for your insightful comment. We realize that our manuscript lacks emphasis on our second contribution (line 84). Our original intention was to state that the sampling process in our proposed ATPO is tree-based, and tree-based sampling can fully leverage the KV cache to save computational costs (we provide a brief theoretical analysis in Appendix A.7). At the same time, in our implementation of ATPO, **the assistant model's answer generation, its interaction with the user model, and the value estimation process by the critic model are all performed asynchronously**, which further improves the throughput of the sampling process, and we have open-sourced our code for reference. To strengthen the emphasis on this contribution in the main text, we have added the following experiments and analyses in our paper:
>     *   In Appendix A.5, we added a statistical analysis of the convergence efficiency and the computational cost during the sampling phase for various algorithms.
>     *   In Appendix A.7, we added a theoretical analysis of how much computational cost can be saved by the tree-based sampling process.
>
>     These analyses help to emphasize our contribution. Furthermore, we also realize that we lacked a clear explanation of the key differences between TreePO and ATPO. Therefore, in the Main Findings subsection of the Results section of our paper, we have further elaborated on their key differences. Specifically, TreePO's original idea was to divide the reasoning chain into multiple segments based on length for tree-based sampling, but this length-based division leads to semantically ambiguous segments. In contrast, **our multi-turn scenario naturally provides semantically clear tree nodes, making it more suitable for tree search**. Moreover, ATPO is also better suited for high-turn dialogue scenarios compared to TreePO (**see the response to Weakness (2)**). Additionally, the analysis in **Appendix A.5** also shows that **ATPO has better convergence efficiency than TreePO**. With these additions, we have further emphasized our contributions and highlighted the key differences from TreePO.
>
> *   **Regarding Weakness (2) [Key differences with TreePO]**: Thank you for your valuable feedback. The performance of ATPO in Table 1 is indeed very close to TreePO in many cases, but the key differences between ATPO and TreePO do not lie there. ATPO has the following advantages over TreePO:
>     *   **Higher convergence efficiency**: As shown in the analysis in Appendix A.5, ATPO converges faster than TreePO. To achieve the same level of performance, ATPO requires fewer computational resources.
>     *   **Better suitability for high-turn dialogue scenarios**: TreePO expands a fixed number of branches each time, causing the number of nodes to grow exponentially with the number of dialogue turns, which quickly exhausts the rollout budget. This results in more diverse data in the earlier dialogue turns but insufficient exploration of later turns. In contrast, for ATPO, as shown in Figures 2(d), (e) in the paper, its adaptive rollout budget allocation mechanism enables the model to better balance the exploration of both shallow and deep nodes, thus making it well-suited for high-turn dialogue scenarios.

---

> > ### Author Response · Authors · 2025-11-18
> >
> > *   **Regarding Questions (1)**:
> >     *   **When the task is not a multiple-choice question**: Although the scenario in our paper is a medical QA task, our proposed algorithm can actually be extended to broader scenarios, such as multi-turn tool use and multi-turn open-ended dialogue. We set the scenario to a medical QA task to be able to quickly provide appropriate rewards to the model based on the correctness of the results. When facing scenarios like multi-turn open-ended dialogue, we can use the "LLM-as-a-Judge" [1] approach to have a more capable model judge the model's output, for example, by evaluating fluency, correctness, and harmlessness, to provide appropriate rewards.
> >     *   **When there are more than 4 options**: The overall process does not need to change, because we never restrict the number of options, but rather instruct the model to choose from the initial information and possible options provided by the user.
> >     *   **When the user does not behave ideally**: We must admit that we assume in the paper that the user can follow instructions: when the assistant asks questions related to the atomic facts provided to the user, the user can answer correctly. This is a very simple task that most models can perform well. We also tested the impact of replacing the user model with others on the test performance. As shown in Appendix A.6, the model achieved a performance level comparable to before, indicating that the assistant model trained with ATPO did not overfit to the user model and has good generalization ability. However, if a user model cannot follow instructions, for example, by consistently failing to find the corresponding atomic facts to respond, this scenario is equivalent to having the model answer directly based on the original information. Even worse, if the user model provides irrelevant or even incorrect interfering information, it will lead to a decrease in test results.
> >
> >     [1] Li, Dawei, et al. "From generation to judgment: Opportunities and challenges of llm-as-a-judge." Proceedings of the 2025 Conference on Empirical Methods in Natural Language Processing. 2025.

---

> > > ### Author Response · Authors · 2025-11-18
> > >
> > > *   **Regarding Questions (2)**:
> > >     *   In certain special cases, pruning can indeed incorrectly remove useful child nodes. This is mainly related to the critic model, the hyperparameter $\tau$, and the number of expanded child nodes $N$. In this paper, $U_1$ primarily reflects the estimation error of the critic model, while $U_2$ reflects both the critic's estimation error and the diversity of the child nodes' values. Therefore, $U_1+U_2$ comprehensively considers trajectories that are useful for optimizing both the critic and policy models. In the early stages, when the critic model is not yet capable, it might incorrectly assign similar value estimates to child nodes, causing useful trajectories to be pruned. We set a 10% probability of skipping pruning in the paper to mitigate this situation. Additionally, setting $\tau$ too high can lead to excessive pruning, causing useful trajectories to be missed. Setting the number of expanded child nodes $N$ too small can lead to missing some useful trajectories due to insufficient exploration and cause large fluctuations in $U_2$. Therefore, considering both efficiency and effectiveness, we set $N$ to 4 in our paper.
> > >     *   Regarding the threshold setting, our objective is to adjust the pruning ratio while minimizing the estimation error of the critic model and enhancing the performance of the policy model. In practice, we observed the approximate range of the value of $U$ and adjusted the threshold within this range. Ultimately, we selected the threshold from the experiment that performed best on the validation set as the final threshold, and then used the model trained with this threshold setting for the final test. It should be noted that hyperparameter selection is strongly dependent on the specific task. In the Future Work section of our paper, we also proposed that future work could explore adaptive hyperparameter selection, for example, by adjusting hyperparameters in real-time based on feedback from the critic loss and pruning rate.
> > >
> > > *   **Regarding Questions (3)**: Regarding lines 255-257, we intended to explain why we use the value $V$ estimated by the critic model in Equation (6) instead of the target value $\hat{V}$ obtained from value traceback. This is because when a node has only one child node, it results in an advantage value of 0 for that node, which prevents the generation of an effective learning gradient. We realize that this description could be confusing. In the revised manuscript, we have improved our description to convey our meaning more accurately.
> > >
> > > *   **Regarding Questions (4)**: Thank you for your valuable feedback. We realize that our paper indeed lacks a detailed description of the atomic fact extraction process. Therefore, in Appendix A.8, we have added the prompt used for extracting atomic facts with Gemini-2.5-Pro. Additionally, we have provided an example of an extracted atomic fact in Appendix A.9 for better reader understanding. Secondly, we found that we had omitted a detailed description of how to evaluate the user's behavior using GPT-4o. Therefore, we have added the corresponding prompt in Appendix A.8 of the paper. We hope these additions will make our paper more complete.
> > >
> > > We hope our responses above have addressed your concerns. We recognize the shortcomings in our manuscript and have made revisions, adding the aforementioned experiments and analyses to make our paper more complete.

---

> > > > ### Comment · Reviewer_Dv8Q · 2025-11-26
> > > >
> > > > Thank you for the detailed clarifications - my concerns are addressed satisfactorily.  I am thus happy to maintain my positive rating.  Good luck!

---

> > > > > ### Author Response · Authors · 2025-11-26
> > > > >
> > > > > We sincerely thank you for taking the time to review our paper and for your positive feedback on our reply. We are glad that our additional experiments and analyses have addressed your concerns. We appreciate your constructive suggestions, which have helped us improve the completeness and clarity of our work.

---

### Author Response · Authors · 2025-11-18

We sincerely thank all reviewers for their insightful comments and valuable suggestions regarding the manuscript. The feedback received has been instrumental in improving the quality and clarity of the research. In response to the comments, detailed point-by-point replies have been provided, and the manuscript has been revised accordingly. In the revised version, all changes are highlighted in **red** for ease of reference.

Specifically, the main changes are as follows:

1. **More in-depth analysis of algorithm efficiency**
    - In Appendix A.5, we have added a detailed comparative analysis of the computational costs of different algorithms, demonstrating ATPO's efficiency advantages, and further highlighting the distinctions from TreePO.
    - In Appendix A.7, we have included a brief theoretical analysis of the computational savings brought by the tree-based approach.

2. **Additional evaluation of dialogue quality**
    - In Appendix A.4, we have added an analysis of changes in the model's effective question rate as ATPO training progresses, providing clearer insights into how ATPO improves dialogue quality.

3. **Strengthened experimental baselines and motivation for reinforcement learning**
    - We have added Gemini-2.5-Pro as an advanced baseline model, making the comparison more comprehensive.
    - To illustrate the limitations of SFT and the necessity of reinforcement learning, we have included a knowledge distillation experiment in Appendix A.3. In this experiment, we fine-tuned the model on expert dialogue data separately generated by GPT-4o and Gemini-2.5-Pro. The results show that the distilled models achieve only very limited improvements, highlighting the shortcomings of SFT and further supporting our proposed goal-driven reinforcement learning method (ATPO) as essential for achieving strong generalization.

4. **Enhanced generalization and robustness analysis**
    - To address concerns about overfitting to the user simulator, we have added a new evaluation experiment in Appendix A.6, where an entirely new user simulator (Llama-3.3-70B-Instruct) was used at test time to replace the training simulator (Qwen3-8B). The results show comparable performance to the previous setting, demonstrating strong generalization capabilities.

5. **Improved clarity and reproducibility**
    - We have refined ambiguous descriptions that could lead to misunderstanding.
    - Further emphasized the key differences with TreePO.
    - To further improve reproducibility, we have included in the appendix A.8 the prompts used for multi-turn dialogue simulation (SFT data construction), atomic facts extraction, and user answer quality monitoring.

We sincerely thank all reviewers for taking the time to review our work once again, and we look forward to any further feedback.

---

### Author Response · Authors · 2025-12-02

Dear AC, SAC, PC, and all reviewers who have participated in the evaluation and discussion of our submission,

We would first like to express our sincere gratitude for the time and effort you have devoted to the review and rebuttal process of our paper. We also deeply regret and are saddened that this round of review has been affected by the recent unexpected incident.

We would like to kindly clarify that, as can be clearly seen from the **timestamps** of comments and replies on OpenReview, **all our discussions, clarifications, and rebuttal exchanges with the reviewers were completed before the system vulnerability was publicly disclosed**, and were conducted in full compliance with the double-blind review policy. During this process, we were fortunate to receive careful, detailed, and constructive feedback from four reviewers, and to engage in thorough and high-quality interactions based on mutual respect.

More specifically:

- The two reviewers (Dv8Q, PxMc) who initially **gave a score of 8 expressed positive and supportive assessments of our work**. After reading our responses and additional clarifications, both reviewers maintained their positive overall judgments.
- For the reviewer (Pcyh) who initially gave a score of 4, we conducted additional experiments and analyses in response to the concerns raised, and we refined and clarified the corresponding parts of the manuscript. After reviewing our replies, this reviewer **increased the score from 4 to 6**.
- For the reviewer (8sJu) who initially gave a score of 2 and raised two main concerns—namely, issues regarding the theoretical derivation and serious worries about potential data leakage—we first provided a detailed clarification in our initial response to address the misunderstanding of the algorithmic mechanism, and we further revised the relevant parts of the manuscript to minimize ambiguity. Building on that, we then designed and added stronger empirical experiments in our second response, which, we believe, fundamentally rule out the possibility that our improvements are due to data leakage. After examining these additional explanations and results, this reviewer also **increased the score from 2 to 6**.
- Based on the above discussions, we systematically incorporated many of the valuable suggestions from the reviewers. On the one hand, we added new experimental results and more detailed analyses to key components such as the theoretical derivation, experimental setup, ablations, and efficiency analysis. On the other hand, we revised and rewrote several descriptions that could potentially cause misunderstandings, in order to improve the overall clarity and readability of the paper. This complete review and rebuttal process has genuinely helped us to significantly refine and strengthen our manuscript.

At the same time, we fully understand and support the decisions made by the ICLR organizers under these difficult circumstances to safeguard the fairness of the review process and the trust of the research community, including rolling back reviewer scores and reassigning ACs. We are well aware that these measures inevitably increase the workload and coordination burden for ACs, SACs, and PCs. We would therefore like to once again extend our sincere thanks and highest respect to all ACs, SACs, PCs, and reviewers who have engaged seriously with our submission. It is precisely because you have continued to uphold rigorous and professional standards, even under such exceptional conditions, that this review process has remained of high academic quality and constructiveness.

Finally, we would like to once again thank you all for your time, patience, and professional guidance on our work.

Sincerely,
The Authors

---

### Meta-Review · Area_Chair_DGee · 2026-01-04

**Summary:**

This paper tackles RL alignment for multi-turn, information-seeking medical dialogue, where the assistant must ask clarifying questions under uncertain user responses before committing to a final diagnosis/answer. The authors formulate the setting as a hierarchical MDP and propose ATPO (Adaptive Tree Policy Optimization), a tree-based RL method that adaptively allocates rollout budget to high-uncertainty states via a composite uncertainty metric and introduces uncertainty-guided pruning plus an asynchronous search and KV-cache reuse architecture to control the cost of tree search.

Across three medical benchmarks, the paper reports consistent improvements over strong baselines (including GRPO and TreePO), with analyses on stability and efficiency; notably, the submission emphasizes that a smaller Qwen3-8B model can be competitive with GPT-4o in their evaluation setting. Reviewers Dv8Q and PxMc view the work as a solid and valuable extension of tree-based RL for long-horizon conversational objectives, supported by detailed experiments and generally strong presentation, recommending acceptance.

Overall, I find the paper to be a meaningful contribution: it addresses an important long-horizon interaction setting, proposes a practically-motivated algorithmic refinement over prior tree-based approaches, and significantly strengthened reporting around compute and efficiency, dialogue-quality analyses, and reproducibility details.

**Reviewer Concerns:**

**Novelty vs. prior TreePO-style methods**

Both supportive reviewers asked the authors to more crisply articulate what is fundamentally new beyond TreePO and to clarify key differences (beyond pruning) and practical parameter choices. The rebuttal and revision add clarifications and expanded discussion/analysis to better position ATPO and its design choices.

**Compute and fairness of comparisons for tree-based RL**

Pcyh’s main concern was that tree methods may consume more compute, and the paper initially lacked adequate reporting of rollout cost or LLM-call complexity, potentially confounding gains. In response, the authors added efficiency analyses comparing time to reach matched accuracy targets and provided additional breakdowns; Pcyh indicated the concerns were largely resolved and updated scores positively.

**Evaluation scope: dialogue quality vs. downstream QA accuracy**

Reviewer 8sJu questioned whether the work truly evaluates multi-turn dialogue quality versus primarily demonstrating improvements on medical QA. The revision adds explicit dialogue-quality analysis to better align method claims with evaluation.

**Theoretical and data-leakage concerns**

8sJu raised questions about aspects of the theoretical derivation and expressed serious concern about potential leakage when using strong LLMs to simulate data. The authors provided detailed clarifications and added additional experiments aimed at ruling out leakage explanations. While this reviewer’s stance in the provided material remains negative, the added evidence and clarifications persuaded the other reviewers that comparisons are reasonably supported.

**Reviewer Scores:**

Reviewer Dv8Q: 8 (accept, good paper); Confidence 4.

Reviewer PxMc: 8 (accept, good paper); Confidence 4.

Reviewer Pcyh: Increased to 6 after rebuttal (main remaining concern was compute-comparison rigor, but largely resolved with added analyses).

Reviewer 8sJu: 2 (reject); Confidence 4, with concerns focused on theory framing, leakage risk, and evaluation alignment.

---

### Decision · Program_Chairs · 2026-01-26

Accept (Poster)